

# Sediment transport modelling in riverine environments: on the importance of grain-size distribution, sediment density and boundary conditions

Jérémy Lepesqueur[1], Renaud Hostache[1], Núria Martínez-Carreras[1], Emmanuelle Montargès-Pelletier[2],
Christophe Hissler[1]

[1]ERIN/LIST, 41 rue du Brill, Belvaux, L4422, Luxembourg
[2]LIEC, CNRS Université de Lorraine, UMR 7360, 54500 Vandœuvre-lès-Nancy, France

*Correspondence to*: J. Lepesqueur (lepesqueur.jeremy@gmail.com)

**Abstract.** Hydromorphodynamic models are powerful tools for predicting the potential mobilization and transport of sediment in river ecosystems. Recent studies even showed that they are able to predict suspended sediment matter concentration in small river systems satisfyingly. However, modelling exercises often neglect suspended sediment properties (e.g. particle site distribution and density), even though such properties are known to directly control the sediment particle dynamics in the water column during rising and flood events. This study has two objectives. On the one hand, it aims to further develop an existing hydromorphodynamic model based on the dynamic coupling of TELEMAC-3D (v7p1) and SISYPHE (v7p1) in order to enable an enhanced parameterization of the sediment grain-size distribution with distributed sediment density. On the other hand, it will evaluate and discuss the added-value of the new development for improving sediment transport and riverbed evolution predictions. To this end, we evaluate the sensitivity of the model to sediment grain-size distribution, sediment density and suspended sediment concentration at the upstream boundary condition. As a test case, the model is used to simulate a flood event in a small-scale river, the Orne River in north-eastern France. The results show substantial discrepancies in bathymetry evolution depending on the model setup. Moreover, the sediment model based on an enhanced sediment grain-size distribution (10 classes) and with distributed sediment density outperforms the model with only two sediment grain-size classes in terms of simulated suspended sediment concentration.

## 1 Introduction

In the last two centuries, many areas have undergone a rather fast demographic, industrial and urban development. This intense land occupancy has affected the quality of surface waters, which become the receptacle of anthropogenic effluents from various origins (Whitman, 1998; Heise and Forstner, 2007; Grabowski et al., 2011). In this context, several rivers in north-eastern France were strongly modified (rectification of river bed, dam building) and received high amounts of industrial and domestic effluents due to former steel-making activities installed near water resources (Kanbar et al., 2017). As a consequence of these past effluent inputs in the river, the riverbeds often remain contaminated, despite part of the settled material having been





dredged and removed from them (Kanbar et al., 2017)., During flood events, the remobilization of these riverbed sediments can strongly impact water and even soil quality (Carter et al., 2006; Hissler and Probst, 2006; Martínez-Carreras et al., 2016). In this context, the composition and status of these contaminated sediments require thorough investigations (SEDNET, 2003) and there is consequently a clear need for predicting the potential resuspension and transport of sediment in these heavily

polluted river systems.

River sediments are aggregates of heterogeneous, composite structures composed of mineral particles of amorphous or poorly crystalline, organic matter, and biological matter (biofilms, bacteria, virus and bio-macromolecules). While fresh sediment deposits are often close to fluid mud, older riverbed sediments are affected by the vertical gradient of consolidation. This vertical differentiation of sediments complicates the modelling of sediment transport, erosion and deposition. Past studies have

shown that hydromorphodynamic models are powerful tools for predicting sediment mobilization and transport, especially in coastal, lacustrine and estuarial and fluvial areas (e.g., Villaret et al., 2013). However, only a few modelling studies applied this type of model to small river systems (e.g., González-Sanchis et al., 2014; Hostache et al., 2014; Hissler et al., 2015). Some promising results were shown with a rather satisfying capability to predict suspended sediment matter concentration. Hydromorphodynamic models often simulate sediment dynamics according to three main processes, namely transport (via

suspended load and bed load), erosion and deposition. Any transport formula assumes that sediment mobilization is triggered when the river bottom shear stress goes beyond a threshold value that depends mainly on grain diameter and sediment density for non-cohesive sediment. Moreover, sediment density strongly influences sediment settling velocity and advection, which govern erosion and deposition via sediment mass balance. In this context, Hostache et al. (2014) highlighted that simulated sediment transport, erosion and deposition are especially sensitive to particle fall velocity, which depends on grain diameter

and sediment density. In addition, those two parameters control the area where a sediment particle is preferentially deposited. Most of the time, hydromorphodynamic models consider sediment as an ensemble of individual spherical particles. For evident reasons, these models do not simulate sediment particles individually, but rather define so-called sediment grain-size classes and simulate sediment transport separately for each class. Belleudy (2000, 2001) and Guillou et al. (2010) emphasized the paramount importance of using enhanced sediment grain size distribution representation to accurately simulate sediment

transport in both coastal and river environments. It has also been shown that uniform grain size for bedload transport can lead to over-prediction in sediment fluxes by a factor of 5 (Durafour et al., 2014). However, the majority of recent studies still consider only few (one or two) sediment grain size classes with uniform density (e.g. Qilong and Toorman, 2015; Hostache et al., 2014) and, in many of them, even a unique median grain size class of sediment is used (García Alba, 2014; Warner et al., 2010). A formal evaluation of model performance when using a larger number of grain size classes and sediment density is

thus still missing.

Here, we further develop an existing hydromorphodynamic model based on the dynamic coupling of TELEMAC-3D and SYSIPHE in order to consider an enhanced sediment grain-size distribution with distributed sediment density. Moreover, this study aims to evaluate and discuss the added value of the new development for improving sediment transport and riverbed evolution predictions. This paper is organized as follows: First, we present the hydromorphodynamic model and the





developments that were made. Second, we describe the study area, the available observation dataset and the experimental design. Next, we present and discuss the results. Finally, we summarize the findings of this study and propose perspectives for future developments in hydromorphodynamic modelling.

## 2 Modelling framework

The proposed modelling framework is based on TELEMAC-MASCARET (Hervouet, 2007). The fluid hydrodynamics are simulated using the TELEMAC-3D model, which solves the Navier-Stokes equations in a hydrostatic mode. The morphodynamic and sediment transport modelling is carried out using the SISYPHE (Villaret, 2010; 2013) model, an additional module of TELEMAC-MASCARET. This modelling framework has the following interests: (i) the two aforementioned models are based on a finite element of unstructured mesh, which is particularly suitable for river and coastal

area modelling as it allows the simulation of complex geometry, and (ii) they can be dynamically coupled. The dynamic coupling of the two models is especially relevant for sediment transport and morphodynamic modelling as it allows, at each simulation time step, to take into account the effect of the riverbed changes on the flow and vice versa. SISYPHE decomposes the dynamic sediment processes into sediment transport, erosion and deposition. Sediment transport is decoupled into the bed load and suspended load and allows sediment concentrations in the water column to be computed.

### 2.1 Friction and bed shear stress

The bed shear stress (τ) is the hydrodynamic variable that mainly controls sediment transport through erosion and deposition (Villaret et al., 2013). TELEMAC-3D uses a roughness coefficient for the bottom energy dissipation by friction. This friction is responsible for the bed shear stress that controls erosion and deposition. In this study, TELEMAC-3D and SISYPHE are coupled dynamically and the friction is calculated based on the Nikuradse law (Nikuradse, 1932). Previous studies on an

estuary system (Lepesqueur, 2009) showed the importance of using spatially distributed friction coefficients instead of a single uniform coefficient in order to obtain more accurate predictions of current velocities and directions, especially in shallow water where the friction is controlled by the apparent roughness of the sediment and the bedforms.

The friction as a function of the bottom sediment grain size (Lepesqueur, 2009), according to the Nikuradse law, is computed as follows:

$$\tau_0 = \rho u_*^2 = \rho \left( \frac{\kappa}{log\left(\frac{30z_1}{k_s}\right)} \right)^2 u_{z_1}^2 \tag{1}$$

In Eq. 1, $\rho$ is the water density, $u_*$ the friction velocity, $z_1$ the distance between the deeper vertical plane from the bed level, $u_{z_1}$ the near bed flow velocity (deeper vertical plane), $\kappa = 0.4$ the von Kármán constant, $k_s \approx 2.5d_{50}$ the Nikuradse bed roughness, and $d_{50}$ the median bottom sediment grain size.





## 2.2 Bed evolution

When TELEMAC-3D and SISYPHE are coupled dynamically the latter computes the bed evolution using the Exner equation (Exner, 1920; 1925) and transmits the bed level state at each time step to the former. The bed evolution is taken into account by the hydrodynamic model to better predict the flow intensity and direction. It is computed based on the divergence of the

bedload flux and the net deposition and erosion due to the suspended sediment transport:

$$(1-n)\frac{\partial Z_f}{\partial t} + \nabla \cdot Q_b + (E-D)_{z=a} = 0 \qquad (2)$$

In Eq. 2, $n$ is the bed sediment porosity, $Z_f$ the bottom elevation, $Q_b$ the bedload flux per unit width, and $E$ and $D$ the erosion and deposit rates at elevation $z = a$, corresponding to the interface between the bedload and suspended load.

## 2.3 Suspended sediment transport

The suspended sediment concentration is computed using the following equation of advection-diffusion:

$$\frac{\partial C}{\partial t} + U\frac{\partial C}{\partial x} + V\frac{\partial C}{\partial y} = \left[\frac{\partial}{\partial x}\left(\gamma_t\frac{\partial C}{\partial x}\right) + \frac{\partial}{\partial y}\left(\gamma_t\frac{\partial C}{\partial y}\right)\right] + \frac{(E-D)_{z=a}}{h} \qquad (3)$$

In Eq. 3, $C$ is the depth-average suspended sediment concentration, $\gamma_t$ is the diffusion coefficient, $U$ and $V$ are the depth-averaged flow velocities in the $x$ and $y$ directions, respectively, and $h$ is the water depth.

## 2.4 Erosion and deposition rates

SISYPHE allows for the consideration of cohesive/non-cohesive sediment mixtures and is able to estimate the evolution of these two sediment types separately. In SISYPHE, the distinction between cohesive (i.e., mud) and non-cohesive sediment is based on the sediment diameter: the sediment is considered cohesive below 63 μm and non-cohesive beyond 63 μm. This is a

relevant point as the processes governing the erosion-deposition of these two types of sediment are markedly different (Villaret et al., 2010). For the cohesive sediment, a uniform suspended mud concentration across the water column is considered. In this case, the Krone (1962) and Partheniades (1965) formulation (see Eqs. 4-5), governs the erosion and deposition rates of cohesive sediment:

$$E = \begin{bmatrix} M * \left(\frac{\tau_0}{\tau_{ce}} - 1\right) & if\ \tau_0 > \tau_{ce} \\ 0 & otherwise \end{bmatrix} \qquad (4)$$

In Eq. 4, $M$ is the Partheniades constant, $\tau_0$ is the shear stress and $\tau_{ce}$ is the critical shear stress.

$$D = \begin{bmatrix} W_s * C * \left(1 - \frac{\tau_0}{\tau_{cd}}\right) & if\ \tau_0 < \tau_{cd} \\ 0 & otherwise \end{bmatrix} \qquad (5)$$

In Eq. 5, $C$ is the suspended mud concentration in the water column and $\tau_{cd}$ the critical constraint of deposition.


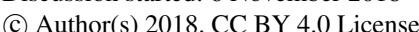


Moreover, SISYPHE is a 2D morphodynamic model, which computes the sediment settling velocity ($W_s$) based on sediment diameter according to Van Rijn (1989)'s formulation:

$$W_s = \begin{bmatrix} \frac{(s-1)gd^2}{18\nu} & if \ d \leq 10^{-4} \\ \frac{10\nu}{d}\left(\sqrt{1 + 0.01\frac{(s-1)gd^3}{18\nu^2}} - 1\right) & if \ 10^{-4} < d \leq 10^{-3} \\ 1.1\sqrt{(s-1)gd} & otherwise \end{bmatrix} \tag{6}$$

In Eq. 6, $s = \rho_s/\rho$ is the sediment relative density, with $\rho_s$ the sediment particle density and $\rho$ the water density, $g$ is the gravitational constant, $\nu$ is the fluid cinematic viscosity and $d$ the sediment particle diameter.

Consequently, the vertical component of the flow velocity is neglected and the particle fall velocity is not directly used in the advection and diffusion of sediment (see Eq. 3). To compensate for this simplification, a vertical Rouse profile of suspended sediment concentration, related to the particle settling velocity in the water column, is assumed for the non-cohesive sediment concentration. This Rouse profile therefore allows the estimation of a so-called reference concentration $C_{ref}$ close to the bottom of the water column that is used for calculating the non-cohesive sediment deposition flux.

Depending on the mud fraction (i.e., ratio between mud and total sediment masses) in the top layer of the river bottom sediment, SISYPHE treats erosion and deposition according to so-called non-cohesive and cohesive regimes. The formulation used for sediment mixture erosion follows the developments of Waeles (2005) that are based on the model proposed by Van Ledden (2001) according to the observations made by Mitchener and Torf (1996), Panagiotopoulus (1997) and Mignot (1989).

The non-cohesive sediment is eroded as pure sand, and is considered a non-cohesive regime if the mass fraction of mud is below 30% and as mud (cohesive regime) if the mass fraction of mud is beyond 50% in the top layer of the river bottom sediment. Where mud is between 30% and 50%, a linear interpolation between the two aforementioned formulations is used. Moreover, in the non-cohesive regime, the non-cohesive sediment is eroded and deposited according to the formulation proposed by Célik and Rodi (1988) using the concept of a so-called equilibrium sediment concentration that is computed using the formulae of Smith and Mc Lean (1977) (see Eqs. 2 and 3):

$$E = \begin{bmatrix} W_s * C_{eq} = W_s * \left(\frac{\gamma_0 T_s}{1+\gamma_0 T_s}\right) & if \ \tau_0 > \tau_{ce} \\ 0 & otherwise \end{bmatrix} with \ T_s = max\begin{pmatrix} \frac{\tau_{skin}-\tau_{ce}}{\tau_{ce}} \\ 0 \end{pmatrix} \tag{7}$$

In Eq. 7, E is the erosion rate, $W_s$ the settling velocity of a sediment particle in the water column, $C_{eq}$ the equilibrium sediment concentration at the bottom of the water column, $C_b$ the sediment bottom concentration ($C_b$=0.65), $\gamma_0$ an empirical coefficient, $T_s$ the normalized excess of shear stress, $\tau_0$ the bottom shear stress, $\tau_{ce}$ the critical erosion shear stress (i.e., the bed shear strength) and $\tau_{skin}$ the shear stress due to skin friction.

Considering a cohesive regime, with a mud fraction beyond 50% in the bottom sediment, the sediment mixture is assumed to be behaving as mud and the bedload is neglected. The erosion rate for the non-cohesive sediment is therefore computed using



the Partheniades (1965) formulation (Eq. 4). Whereas the erosion rate of the non-cohesive sediment is treated differently depending on the mud fraction in the bottom sediment, the deposition rate of the non-cohesive sediment is invariably computed using:

$$D = W_s * C_{ref} \tag{8}$$

In Eq. 8, $D$ is the deposition rate and $C_{ref}$ the reference sediment concentration at the bottom of the water column.

In this case, the erosion flux is assumed to be initiated only if the bottom shear stress becomes higher than the threshold value (i.e., the critical Shields number). When the bed shear stress is below the critical Shields number, no motion occurs. On the contrary, if the bottom shear stress exceeds the critical Shields number, the sediment starts moving. Shields (1936) was the

first author to lay stress on the initiation of sediment transport as a threshold process. For cohesive sediment, this threshold corresponds to the critical shear stress of erosion that is an intrinsic property of the mud and can be assessed using a scissometer. For the non-cohesive sediment, the threshold for initiating motion is more empirical (Shields, 1936). In this study, the Shields' parameter is introduced:

$$\theta_s = \frac{\tau_0}{(\rho_s - \rho)gd} \tag{9}$$

In Eq. 9, $\theta_s$ is the Shields' parameter. The erosion of non-cohesive sediment is initiated if the Shields' parameter exceeds a so-called critical Shields' number (Soulsby and Whitehouse, 1997), defined as:

$$\theta_c = \frac{\tau_{ce}}{(\rho_s - \rho)gd} = \frac{0.3}{1 + 1.2d_*} + 0.055(1 - e^{-0.02d_*}) \qquad with \ d_* = d\left[\frac{g(s-1)}{v^2}\right]^{1/3} \tag{10}$$

In Eq. 10, $\theta_c$ is the critical Shields number and $d_*$ the dimensionless sediment particle diameter.

The threshold for the initial motion of non-cohesive sediment is based on the ratio of a critical bed shear stress and the submerged grain weight. Many studies proposed a less empirical parameterization based on the weight and the (angular) surface of the sediment grain but eventually showed results quite similar to those obtained when using the original Shields curve (Zanke, 2003; Miedima, 2010). Consequently, one can argue that the Shields curve can still be considered a good means for assessing the criterion for the mobility of homogeneous non-cohesive sediment. Many studies proposed a modulation of

the Shields curve based on experiments with heterogeneous sediments (e.g., Zanke, 2003). In this study, the formulation proposed by Soulsby and Whitehouse (1997) is used to calculate the Shields parameter. This is derived from the initial Shields curve with a better fit at a low Reynolds number, therefore improving the accuracy for smaller diameters (see Eq. 4).

## 2.5 Bedload flux

As mentioned in Section 2.4, the bedload flux is neglected in a cohesive regime. However, in a non-cohesive regime, the formulation of Meyer-Peter-Müller is used to compute the bedload flux:


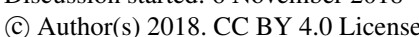


$$Q_b = \begin{bmatrix} \alpha_{mpm}(\theta_c - \theta_s)^{3/2}\sqrt{g(s-1)d} & if & \theta_s > \theta_c \\ 0 \; otherwise \end{bmatrix} \qquad (11)$$

In Eq. 11, $Q_b$ is the bedload flux and $\alpha_{mpm}$ the Meyer-Peter-Müller coefficient. The excess of bed shear stress responsible for the sediment mobilization is the difference between the skin friction (i.e. Shields parameter) and the critical bed shear stress calculated using the critical Shields number.

**2.6 Sediment grain-size distribution and bottom sediment composition**

In its original version, SISYPHE is limited to two-class sediment mixtures (cohesive and non-cohesive sediment). To circumvent this limitation, we enable SISYPHE to run simulations for a 10-class sediment mixture: three classes of cohesive sediment and seven classes of non-cohesive sediment. As in the initial version of SISYPHE, each class is defined by a median grain diameter and a nominal density in this study. Each sediment class can be treated separately and its characteristics (the Shields number and the settling velocity) and the nominal erosion, deposition and transport rates are computed separately for each class. Finally, the global sediment erosion, deposition and transport rates are estimated by summing the sediment class nominal contributions.

Over the model domain, the bottom sediment mixture is defined based on the volumetric fraction of each sediment class. Moreover, the bottom sediment is stratified in ten layers defined by their respective thickness as a function of the median sediment grain size:

$$ES(i) = i^2 * d_{50}(i) \qquad (12)$$

In Eq. 12, $ES(i)$ is the thickness of the layer $i$ and $d_{50}$ the median grain size. The top layer defines the active layer. The second layer starts to be eroded when the coarser sediment of the first layer has been totally eroded, otherwise the flux of erosion of finest particles is limited to the first active layer.

**3 Study area, available data, model set up and experimental design**

**3.1 Study area**

The Orne River, located in north-eastern France, drains around 1270 km$^2$ and flows into the Moselle River. Since 2014, the maximum discharge that has been recorded is higher than 200 m$^3$/s, corresponding to a flood return period of approximately ten years. At low flow, the turbidity of the Orne River is particularly low (< 5 NTU). We selected a 4 km-long control section (Fig. 1) for this modelling exercise of suspended sediment transport. In the area of interest, the riverbed has an average width of 30 m and an average slope of 0.1%. The modelled reach is composed of two large meanders. Its downstream boundary is equipped with a dam. The streambed is mainly composed of pebbles, coarse gravel, sand and a small silt portion. The



riverbanks are mainly composed of a sand-mud mixture with varying contents of mud and are covered by dense vegetation. At some locations, the riverbanks are made of concrete or silted-up rockfills.

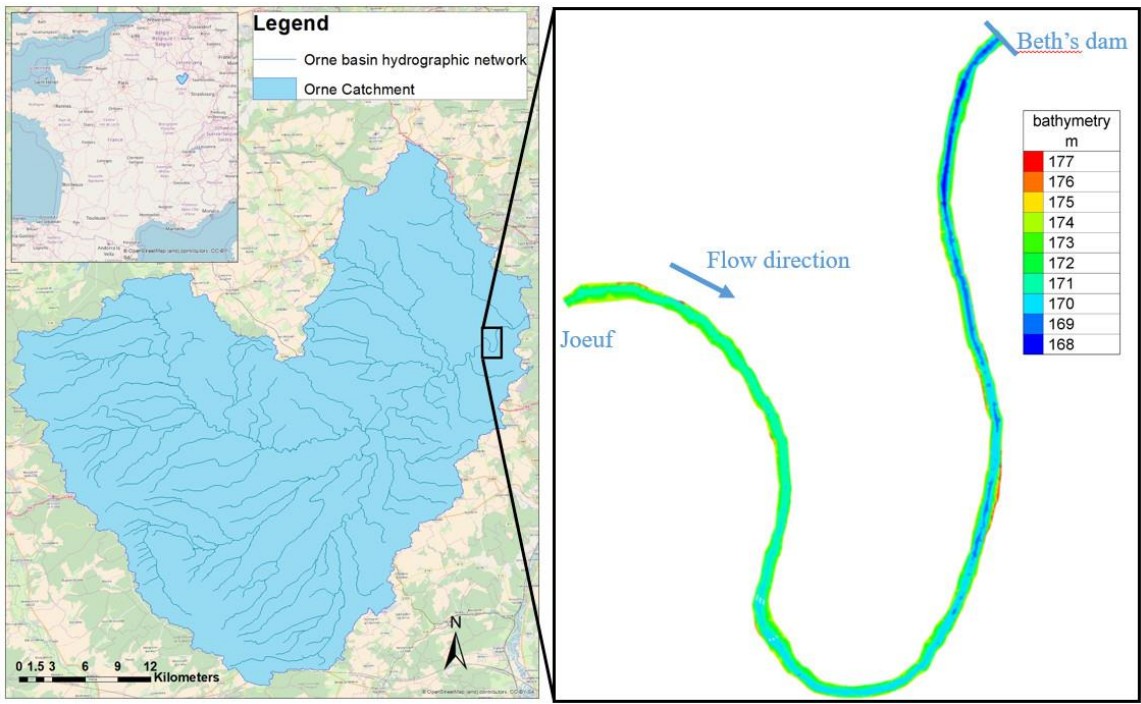

**Figure 1: Study area and model domain (4 km-long control section of the Orne River)**

## 3.2 Available data

Since January 2014, monitoring efforts have been concentrated on continuously recording streamflow and water turbidity as a proxy of suspended sediment concentration. Moreover, bathymetry (i.e. riverbed elevation) and sediment deposition were measured more episodically at selected locations on the riverbanks and the riverbed. The continuous data used in this study were acquired during a moderate-magnitude flood event that occurred in March 2017. During this event, a peak discharge of 45 $m^3.s^{-1}$ was recorded and the turbidity did not exceed 150 NTU (Fig. 2).

### 3.2.1 Suspended Sediment Concentration (SSC)

SSC is generally measured punctually whereas models require continuous input data time series. In this context, turbidity data is often recognized as a good proxy for estimating continuous the time series of SSC (Martínez-Carreras et al., 2016). In this study, the turbidity is monitored every 5 minutes at the downstream boundary using an YSI 600 OMS turbidimeter. During the flood event, turbidity values ranged from 0 to 150 NTU. These measurements are used to calibrate the relationship between




turbidity and SSC. The polynomial regression between the two datasets (e.g. Versini et al., 2015) exhibits a Pearson's correlation coefficient of 0.968 and a residual mean of 1.44 mg·L$^{-1}$ (Fig. 2). The calculated SSC is compared to the observation in Fig. 3.

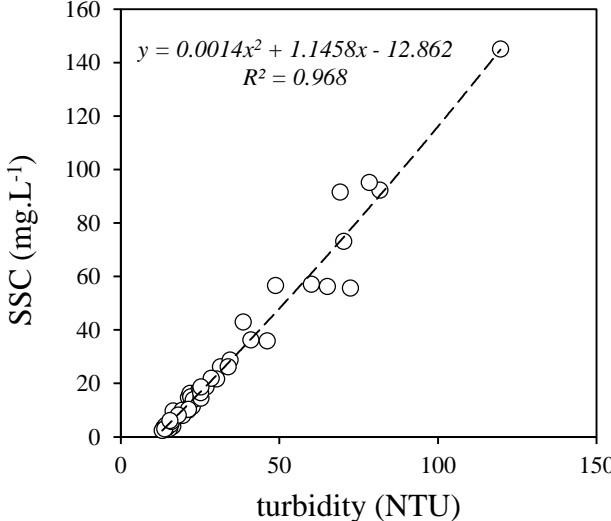

$$y = 0.0014x^2 + 1.1458x - 12.862$$
$$R^2 = 0.968$$

**Figure 2: Relationship between the river water turbidity and the suspended sediment concentration (SSC) measured at the downstream boundary in the Orne River section studied.**

Water samples were automatically collected every 6 hours using ISCO© automatic samplers at the upstream and downstream boundaries. The similarities observed and SSC estimated at various locations (Fig. 3) indicate that the sampling frequency is

10 sufficient to capture the suspended sediment dynamics in the river section during this event. As a consequence, the ISCO sample-derived SSC at the upstream location is used as an upstream forcing of the model. The SSC was measured by filtering about 1L of river water through 1.2 μm Whatman GF/C glass fibre filters by means of a Millipore vacuum pump. All filters were previously dried at 105 °C for 24 hours, cooled in a desiccator and weighted. After filtration, the filters were dried again at 105 °C and reweighted. The differences between weightings provided the total amount of sediment retained in the filters.

15 We calculated the SSC by dividing the total amount of sediment retained in the filters by the volume of the filtered samples.





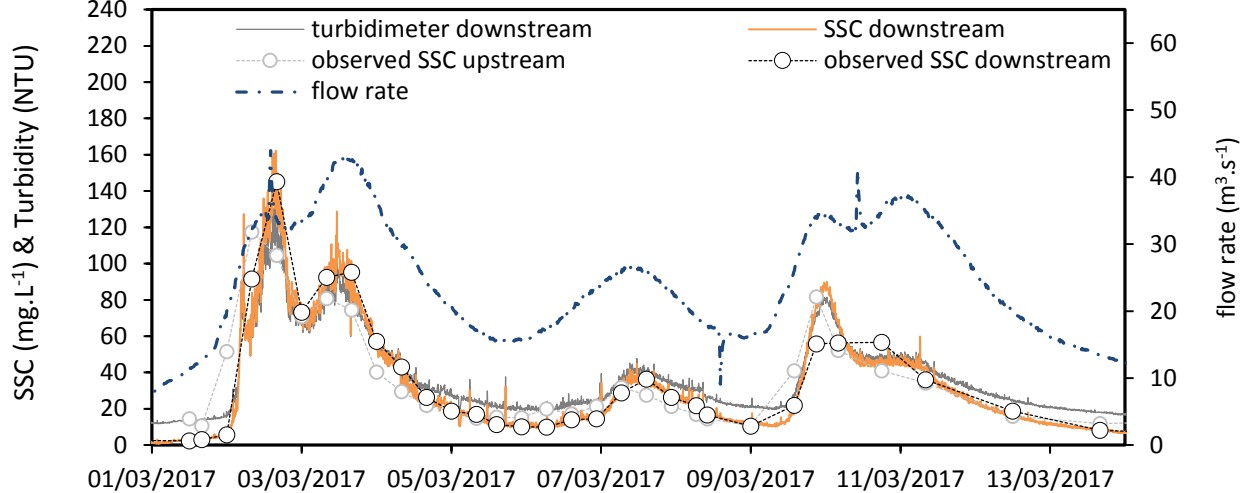

**Figure 3: Times series of flow rate, turbidity and calculated SSC. SSC observed at upstream and downstream boundaries of the model domain are also plotted for comparison.**

### 3.2.2 Sediment grain size distribution

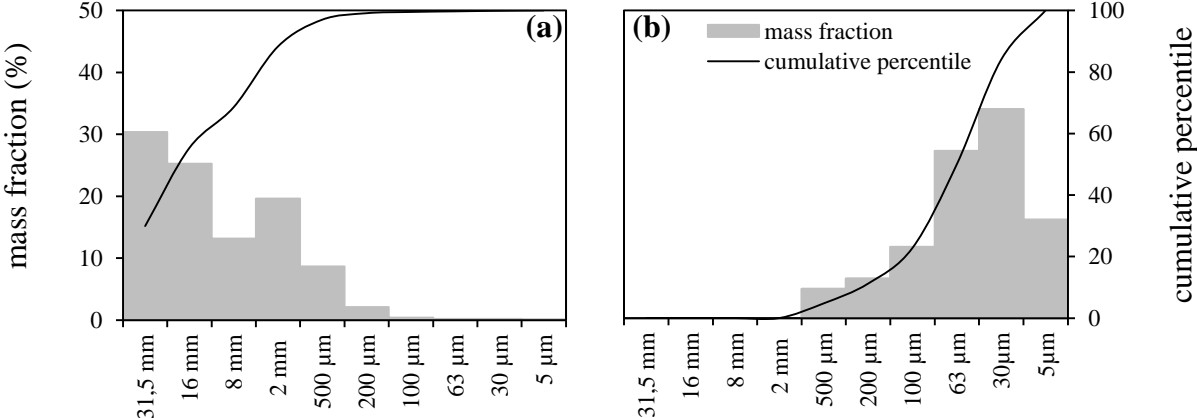

**Figure 4: Sediment grain size distributions estimated from the Orne River sediment samples collected in (a) the riverbed and (b) the riverbanks.**

We estimated the grain size distributions showed in Fig. 4 by sieving dried sediment samples collected in three different areas of the river section. Due to deep water at the downstream end of the river section caused by the dam, it was technically impossible to collect riverbed sediments in this part of the river. Moreover, as an extensive sampling of sediments along the river was not feasible, we assume, as in the initial conditions in the modelling exercise, that the riverbed and riverbank sediment grain-size distributions are homogeneous along the river reach. These initial sediment grain size distributions are actually estimated by averaging the three sediment samples.




### 3.2.3 Sediment density

In sediment transport modelling, the density of the sediment is usually set to 2600 kg.m$^{-3}$ (Van Rijn, 1984). Here, we suggest considering a measured sediment density for each sediment class. To this end, we measured the variation of water volume in a 400 mL graduated flask while pouring a predefined mass of sediment into the water. The density measurements exhibit a

spread of 1000 kg.m$^{-3}$ and an average value of 2300 kg.m$^{-3}$. The minimum density is 1800 kg.m$^{-3}$ for the 63 µm sediment class (Fig. 5).

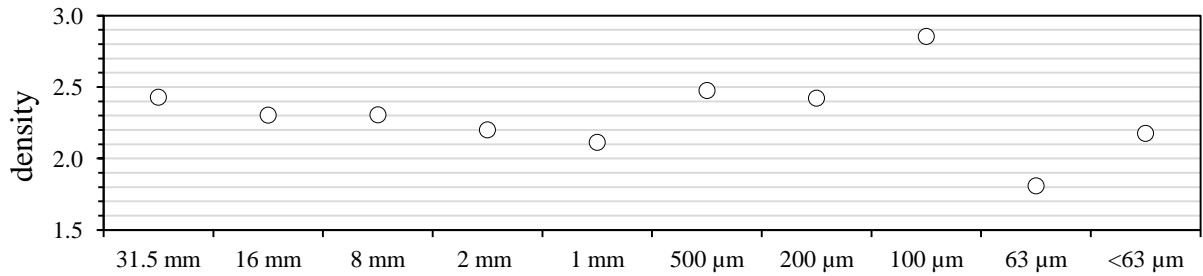

**Figure 5: The Orne River bottom sediment density measured for each of the 10 sediment size classes.**

### 3.2.4 Riverbed bathymetry

The bathymetry of the riverbed and the lower part of the banks was carried out during two field campaigns using a Differential GNSS system (vertical accuracy c.a. 1 mm) coupled with an echo-sounder (vertical accuracy c.a. 1 mm). The ground elevation of the upper parts of the banks was measured using a Differential GNSS system (vertical accuracy c.a. 1 mm) and a total station (vertical accuracy c.a. 1 cm) when the GNSS signal was not accurate enough due to the dense vegetation cover. These

campaigns allowed us to measure riverbed elevation along the river cross section every c.a. 100 m.

### 3.3 Model setup and experimental design

Particularly well-adapted to simulate river hydrodynamics, TELEMAC-MASCARET is based on a finite element unstructured mesh allowing for representing complex geometry (Hostache et al., 2014). For the study area, the unstructured mesh is composed of 16492 nodes distanced from 7 m up to 25 m. It was generated with POLYMESH@ (developed by A. Roland,

T.U. Darmstadt) using a criterion on the bathymetry. The six bridge piles lying in the domain are represented in the model geometry. The riverbed and riverbank sediments are defined with two distinct grain-size distributions (Fig. 4).

Four model configurations have been designed in order to assess the sensitivity of the model predictions to the sediment grain-size distribution, the sediment particle density and the boundary conditions (Tab. 1). The SISYPHE and TELEMAC-3D parameter values remain identical for the four different modelling configurations. The SSC distribution is assumed to be equal

to the distribution of the erosion fluxes of each class at the boundary conditions. The settling velocity is calculated for each




sediment class using the experimental sediment density values (Eq. 7 and Fig. 5). Moreover, due to the presence of vegetation on the riverbanks, the corresponding apparent roughness is fixed to 4 cm for the four modelling setups.

**Table 1: Model configurations used in this study**

| Model configuration name | Suspended Sediment classes | Bottom Sediment classes | Density |
|---|---|---|---|
| 2CL | 2 | 2 | variable per class |
| 10CL | 10 | 10 | variable per class |
| 10CLD | 10 | 10 | 2600 kg m$^{-3}$ |
| 10CL1CS | 2 | 10 | variable per class |

The first configuration (2CL) corresponds to the standard SYSIPHE configuration, which only considers two classes of particle sizes with distinct densities. The second configuration (10CL) considers a riverbed composed of bottom sediment with ten classes with distinct density values (Fig. 5) and an input suspended sediment concentration, at the upstream boundary condition, distributed over the same ten classes. The third configuration (10CLD) differs from configuration 10CL in terms of sediment density: the ten sediment classes have the same "standard" density value (i.e. 2600 kg.m$^{-3}$). Configuration 10CLD uses a density value of 2600 kg m$^{-3}$. Note that the "standard" density value is higher than the ones we measured in the laboratory for all the sediment classes except for the 100 μm (2850 kg m$^{-3}$; Fig. 5). The last configuration (10CL1CS) is identical to configuration 10CL except that the input suspended sediment concentration is imposed only on the sediment with the smallest particle size (<5μm).

## 4 Results and discussion

This section presents, evaluates and discusses the results obtained based on the four model configurations (Table 1). In particular, it aims to evaluate the influence of the sediment size distribution, sediment density and boundary condition representation on the simulated SSC and the bed evolution, respectively. To carry out this evaluation, the simulated SSC at the downstream boundary of the model domain is first compared with the corresponding observed data.

### 4.1 Evaluation of the simulated SSC

### 4.1.1 Influence of the sediment grain size distribution

The 2CL configuration required some additional effort for the model initialization and spin-up. Indeed, without a numerical adjustment of the initial bathymetry, the 2CL configuration was unable to yield a satisfying fit with observed SSC data as spurious fluxes of SSC appeared (Fig. 6a). Some authors (e.g., Waeles, 2005) reported the need for long simulations (up to one year) in order to obtain a satisfying initial state of the bathymetry and the sediment repartition. In our study, we successively simulated the same event several times. Five iterations (referred as 2CL1, 2CL2, …, 2CL5) were necessary in order to stabilize





the initial bathymetry and avoid a systematic overestimation of the first SSC peaks (Fig. 6a and 6b). We took the fifth run of the 2CL configuration (i.e., 2CL5) as a reference for the discussion as it yields the best results in terms of simulated SSC. Model initialization and spin-up were not necessary for the other configurations, namely 10CL, 10CLD and 10CL1CS.

Table 2 clearly shows that better model performances are obtained when using a larger number of grain-size fractions/classes.

Indeed, not only are the error metrics substantially reduced in the 10CL configuration (in comparison to the 2CL5 configuration), but also the Pearson's correlation coefficient and the Nash–Sutcliffe efficiency (NSE) increase significantly. Moreover, as shown in Fig. 6, the 2CL5 configuration tends to overestimate the first SSC peak (maximum absolute error: 118 mg·L$^{-1}$) and underestimate SSC for the rest of the simulation (mean error: -7 mg·L$^{-1}$). This highlights the limitations of a 2-class model that is not able to correctly predict SSC both at rather low and high flows. On the contrary, the 10CL configuration

is able to accurately capture SSCs as the mean and the maximum errors are 1.6 mg·L$^{-1}$ and -45 mg·L$^{-1}$, respectively.

### 4.1.2 Influence of the suspended sediment density

As a reminder, in the 10CL model configuration, we use distinct densities for each class of sediment (Fig. 5), whereas we use a unique value of density in 10CLD (2600 kg·m$^{-3}$). During the simulated event, the contribution of the non-cohesive sediment to the SSC is very small (in the order of 1 mg·L$^{-1}$ at maximum). Both configurations accurately reproduce the observed SSC

(Fig. 6c). However, the 10CL configuration slightly outperforms 10CLD (Table 2) as the peaks of SSC are better predicted in the 10CL configuration than in the 10CLD configuration: the first and the last peak of SSC during the event exhibits a difference of 10 mg·L$^{-1}$ between the two models, which is not negligible as it represents for instance 10% of SSC during the last peak. The fall velocities are directly linked to the density (Eq. 6). As a result, overestimating sediment density can significantly reduce simulated SSC. Although the effect of sediment density on model results is slightly limited in our

experiment, mainly because the simulated event is of a rather moderate magnitude, one could expect a higher sensitivity for larger flood events.





**Figure 6: Simulated and observed suspended sediment concentration time series at the downstream boundary for configurations: a) 2CL (1ˢᵗ run); b) 2CL (4ᵗʰ and 5ᵗʰ runs); c) 10CL, 10CLD and 10CL1CS.**





**Table 2: Model performances computed for a 14-day simulation period (1-14 March 2017)**

| Model Configuration | Mean error (mg·L⁻¹) | Max error (mg·L⁻¹) | RMSE (mg·L⁻¹) | NRMSE % | CORR | NSE |
|---|---|---|---|---|---|---|
| **2CL** | -7.86 | 118.99 | 14.74 | 37.67 | 0.89 | 0.72 |
| **10CL** | 1.60 | -45.89 | 8.23 | 21.04 | 0.95 | 0.91 |
| **10CLD** | 0.84 | -49.59 | 8.57 | 21.91 | 0.95 | 0.90 |
| **10CL1CS** | 5.22 | -34.54 | 9.14 | 23.36 | 0.96 | 0.89 |

### 4.1.3 Influence of the suspended sediment size distribution imposed at the upstream boundary

In the10CL1CS configuration, the simulated SSC is generally higher than in 10CL (Fig. 6c). This is mainly due to the way the upstream SSC is imposed in the 10CL configuration. Indeed, as the input SSC is distributed over 10 classes, coarser particles tend to settle more rapidly and the predicted downstream SSC is then lower than in the 10CL1CS configuration. Overall, the error metrics and the skill scores reported in Table 2 show that the 10CL configuration slightly outperforms the 10CL1CS configuration as errors are lower and NSE is slightly higher.

Overall, due to the rather moderate magnitude of the simulated event, the main processes controlling simulated downstream SSC appear to be advection and diffusion. To further investigate this, Fig. 7 shows the cumulative (starting from larger grain size) distribution of SSCs per sediment class simulated at the downstream boundary by the 10CL and 10CL1CS configurations. As can be seen in this figure, the contribution of non-cohesive sediments to the overall SSC is rather limited (in the order of 1 mg·L⁻¹ at maximum). Indeed, it only contains the 100 μm sediment class for both models. However, as visible in Fig. 7b,

erosion within the domain contributes slightly to the SSC as 63 and 30 μm-sediment classes are transported in suspension in the10CL1CS configuration whereas this configuration imposes SSC input only on the finest sediment class (5 μm). Moreover, as the two configurations considered produce markedly different results in terms of suspended sediment size distribution (Fig. 7), the way the upstream boundary condition is defined is shown to have a significant importance, especially on the advection and diffusion processes. We hypothesize that the difference between the SSC simulated with the two different configurations

would be even larger when simulating higher magnitude flood events as the coarser and heaviest particles are more subject to sedimentation. Moreover, the dam affects circulation at the study site, reducing current velocity. Hence, the heaviest particles that can be transported at the upstream boundary, if the current velocity is high enough, might not reach the downstream part of the river due to the influence of the dam.





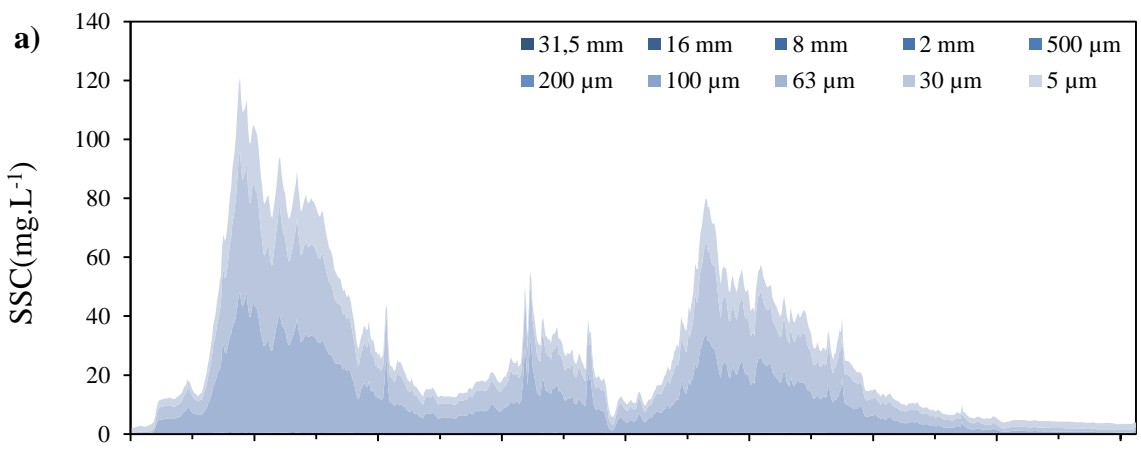

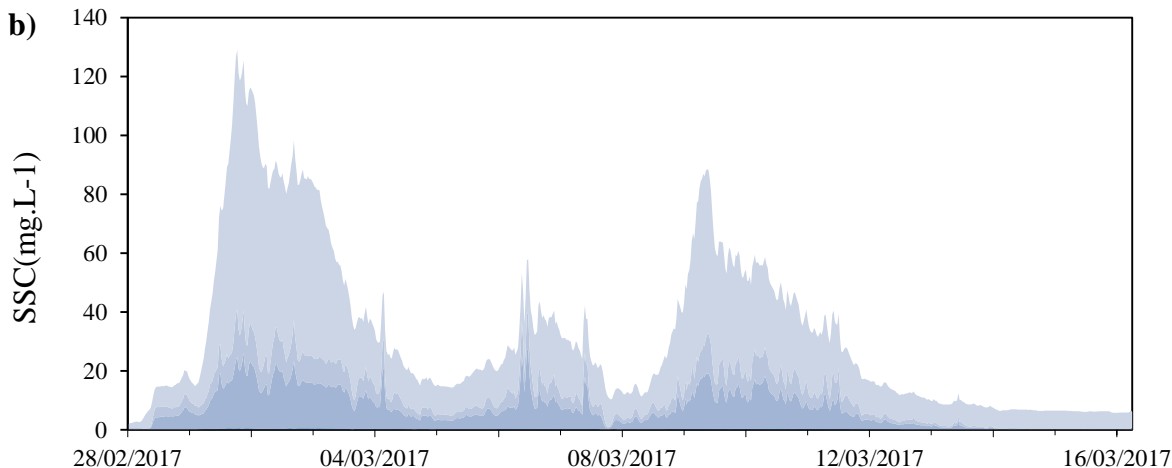

**Figure 7: Downstream suspended sediment grain size cumulative distribution simulated by model configurations (a) 10CL and (b) 10CL1CS.**

## 4.2 Cross-comparison of simulated riverbed evolution

Comparing simulated bathymetry evolution maps showing changes in riverbed elevation is not straightforward for a moderate magnitude event on a small river, especially because the evolutions are rather limited and local. To facilitate such a comparison, the evolutions of the riverbed elevation simulated by the various model configurations are compared via scatter plots (Fig. 8)

10 using the 10CL configuration as a reference. Using bathymetry evolution instead of bathymetry itself not only allows a differentiation between erosion and deposition, but also an assessment of the thickness of deposited and eroded material. The bathymetry evolutions simulated by the 10CL configuration are separated as follows: erosion area (elevation change<-5 mm), stable area (elevation change in [-5mm:5mm]) and deposition area (elevation change > 5mm).




### 4.2.1 Influence of the sediment grain-size distribution

The comparison between evolutions obtained with the 2CL and 10CL configurations shows very low correlation coefficients (0.17 and 0.02 for the erosion and deposition, respectively). Moreover, stable areas in the 10CL configuration are unstable in the 2CL configuration (-0.11 of correlation). These substantial differences between the two configurations confirm that the number of sediment size classes implemented in the model plays a central role in the simulation of erosion/deposition processes. Overall, riverbed evolutions simulated by the 2CL model configuration are almost inexistent. This is mainly due to the model spin-up (see Section 3.1) that was necessary for stabilizing the bathymetry. The simplified sediment size distribution (two classes) artificially amplifies the availability of the finest sediment class. This leads to a washout of this class during the spin-up simulation and at the beginning of the event simulation.

### 4.2.2 Influence of the suspended sediment density

The middle scatter plot in Fig. 8 shows a good correlation between riverbed evolutions simulated by the 10CL and 10CLD configurations. The correlation coefficients computed on erosion and deposition areas are high with respective values of 0.97 and 0.92. Therefore, we argue that the sediment density has some influence on the morphological changes occurring, especially for the deposition. Nevertheless, the correlation of the deposit should decrease as the flow rate increases, especially for more intense flood events. The more SSC is composed of different classes, the more the density would have an effect on deposition, as the density is directly linked to the fall velocity and the fall velocity induces the location and amount of deposition.

### 4.2.3 Influence of the suspended sediment size distribution at the upstream boundary

The right scatter plot in Fig. 8 shows a good correlation between the10CL1CS and 10CL configurations in terms of deposition and erosion areas, with respective values of 0.99 and 0.98. As argued previously, the differences in bathymetry evolution between the two configurations (10CL and 10CL1CS) would likely be more important in the event of a higher flow rate.

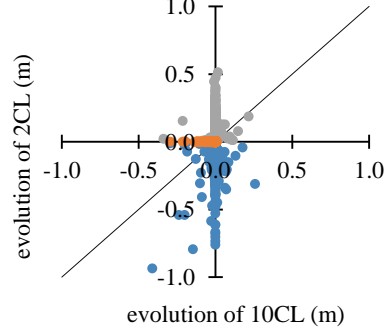
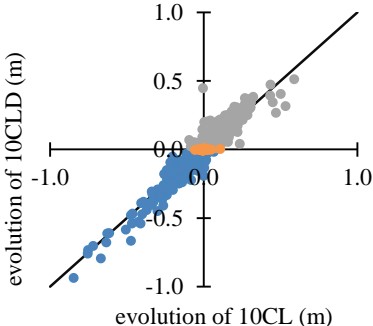
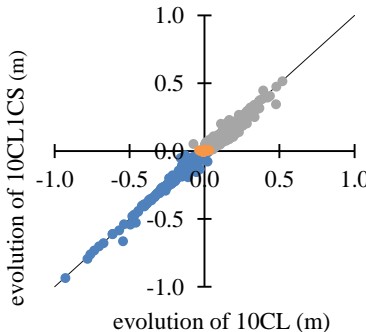



**Figure 8: Cross-comparison of bed elevation evolutions (elevation final-initial) simulated for the 2CL, 10CL1CS and 10CLD configurations using the model configuration 10CL as a reference. The colours correspond to the 10CL's bathymetry evolution: the grey is the deposition, the orange is the stable bathymetry and the blue is the erosion.**

## 4.3 Cross-comparison of simulated bottom sediment median grain-size evolution

The median grain size of the riverbed sediment at the end of the simulation is analysed by cross-comparing the evolution (final-initial) of the median grain size (D50) at each grid node. The 10CL configuration is taken as the reference (vertical axis).

### 4.3.1 Influence of grain-size distribution of the suspended sediment

Fig. 9 shows that there is very limited correlation between the D50 evolutions when using the 10CL and the 2CL configurations (correlation coefficient of -0.05). The median evolution of the D50 in the whole area when using the 2CL configuration is about 70 μm: the fine particles tend to leave the domain and the D50 increases. On the contrary, the median evolution of the D50 when using the 10CL configuration is null: there is an equilibrium of the D50 in the domain, indicating that the grain-size sorting during the event does not change the D50 over the domain.

### 4.3.2 Influence of the suspended sediment density

The cross-comparison (Fig. 9) shows a weak correlation between the evolution of the D50 when using the 10CL and 10CLD configurations (correlation coefficient of 0.32). The bottom sediment distribution simulated is hence strongly impacted by the sediment density. The distribution of the points in the cross-comparison is less spread out on the horizontal axis, suggesting that the distribution of the sediments is more stable for the 10CLD configuration. The common value of 2600 kg·m⁻³, which is higher than the mean sediment density measured in our field study, tends to stabilize sediments.

### 4.3.3 Influence of the grain size distribution of suspended sediment at the upstream boundary

The cross comparison (Fig. 9) shows a high correlation between the evolutions of the D50 in the 10CL and 10CL1CS configurations (correlation coefficient of 0.87). The median (over the whole area) evolution of the bottom sediment D50 in the 10CL1CS configuration is null as well, suggesting that there is an equilibrium of the D50 all over the domain, as for the 10CL configuration. This particular flood event was of low intensity. Consequently, the fraction of fine sand imposed at the boundary condition of the 10CL configuration was negligible and the suspended sediment was distributed over the three cohesive sediment classes. Hence, the comparison of 10CL and 10CL1CS is highly limited by the moderate magnitude of the event.





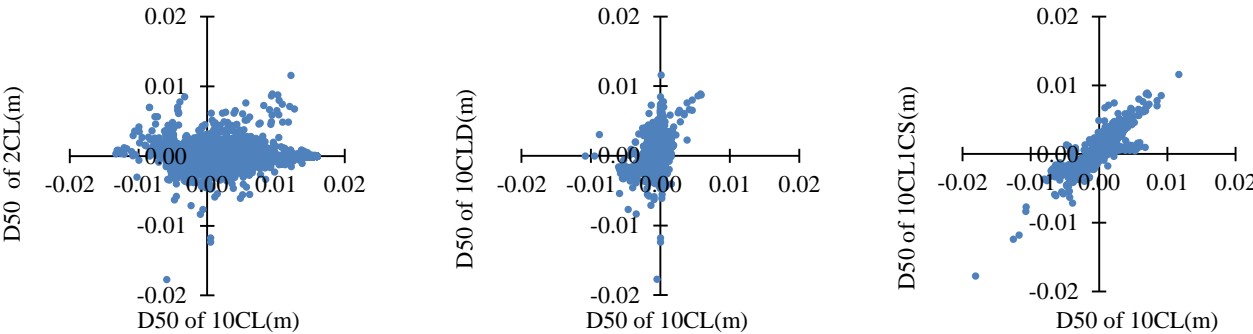

**Figure 9: Cross-comparison of riverbed sediment median grain-size evolution (final-initial) simulated using the 2CL, 10CL1CS and 10CLD configurations using the 10CL configuration as a reference model .**

## 5 Conclusion

This study evaluates the influence of the sediment grain-size distribution, the sediment density and the boundary condition representation on sediment transport/morphodynamic modelling. In this context, the SYSIPHE model has been further developed in order to integrate ten classes of sediment (mixture of sand and mud) with individual sediment densities (two sediment classes are implemented in the standard version). The physical parameterization has also been rewritten, based on the parameterization proposed by Lepesqueur (2009), and has been adapted to the last release of SISYPHE (i.e. from version V5P8 to V7P7). The enhanced SISYPHE model is evaluated using a moderate magnitude flood event of a small river (the Orne River, north-eastern France) as a test case.

The following conclusions are drawn from this study:

1. The simulated suspended sediment concentration (SSC) is markedly improved if the model takes into account 10 sediment classes instead of 2. The RMSE on SSC is reduced by a factor of 2 with 10 sediment classes. The simplified model, including only 2 sediment classes appeared to simulate spurious sediment fluxes. Considering 2 or 10 classes of sediment in the model results in markedly different erosion/deposition areas.

2. The sediment density is, albeit to a smaller extent, substantially influencing model results. Using measured sediment densities (individually for each sediment class) instead of a standard uniform value (i.e., 2600 kg·m$^{-3}$) allowed for a slight gain in model performance on simulated SSC. The area of erosion/deposition slightly changed when using measured densities.

3. The way the input SSC is imposed at the upstream boundary also plays a role, albeit a limited one in this particular flood event, in the model performance. However, the influence on erosion/deposition is not significant.



## 6 Future scope

In the proposed modelling framework with an improved representation of sediment properties (number of sediment classes, densities and input SSC discretized over the classes of sediment), the numerical results proved to be more accurate. However, improvements are of course still needed and this brings forward further processes that could be introduced in a future modelling

framework:

1.  The erosion and deposition laws should also take into account the overall interactions between the different sediment classes at the bottom. Indeed, many mechanisms (e.g. compaction of non-cohesive sediment, armouring, hiding/exposure, filtration of fine particles by coarser sediment and lubrication induced by fine particles on coarser sediment) can be responsible for heterogeneity in the bottom sediments, which can either stabilize or destabilize the
sediment leading to a reduction or increase of the erosion fluxes (Swidersky, 1976; Starck, 2014; Egiazaroff, 1965; Ashida, 1973; Karim, 1982; Brunke, 1999; Herzig et al., 1970; Barry, 2006; Arthur et al., 1980; Widdows et al., 2000; Le Hir et al., 2007) . Integrating these mechanisms in morphodynamic modelling could contribute to further improving sediment transport predictions.

2.  In this study, the input SSC is numerically distributed over the sediment classes based on the riverbed sediment class
distribution. However, it would be certainly beneficial to measure the SSC per sediment class directly to avoid introducing a bias and impose a more realistic sediment flux.

3.  Small particles in suspension can aggregate each other thereby creating flocs (Parker, 1972; Van der Lee, 2009). The flocculation process plays a role in sediment transport as the density and the shape of flocs is different from those of individual sediment particles. Their displacement in the water column is different from that of isolated sediment
particles as a result of their different settling velocities and diffusion properties. As a consequence, taking into account flocculation could also help improve sediment transport modelling in the future.

## Acknowledgments

This study is part of the MOBISED project co-funded by the Luxembourg National Research Fund (FNR) and the
French National Research Agency (ANR) in the framework of the FNR/INTER-ANR research programme (Contract No. INTER/ANR/13/9441502). We would like to thank Jean-François Iffly, Jérôme Juilleret, Luc Manceau and Cyrille Taillez for the maintenance of field equipment and the accurate field data acquisition, and Claire Delus and Benoît Losson for the constructive scientific discussions related to hydrological and sedimentary issues in the Orne River basin.




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
