# Peer review of "Sediment transport modelling in riverine environments: on the importance of grain-size distribution, sediment density and suspended sediment concentrations at the upstream boundary"

_Hydrology and Earth System Sciences, 2018_

## Short Comment (SC1) · 12 Nov 2018

The paper was elaborated based by the use of SISYPHE. Within this code, the determination of settling velocities obviously is cited after v. Rijn (Hand book of Sediment Transport by currents and waves. Delft Hydraulics, Rep. H461, 1989).

This formula, Eq. 6 in the paper under discussion has a spelling error. The number '18' in the denominator of the middle equation must be deleted.

The area-wise subdivision is unnecessary (saves at most minimal computation time).

[Figure]

The middle equation also includes the solution "otherwise" and the solution for "d $\leq$10-4".

It should also be noted that the grain size data such as "d $\leq$10-4" only concern sands. However, the equation applies to any sediment densities in which the grain size assignments of Eqn. 6 then would not apply.

Apart from that, Eqn. 6 with v. Rijn cited as its author, is not correctly cited. Correct citation is

Zanke, U. (1977): Berechnung der Sinkgeschwindigkeiten von Sedimenten (Determination of settling velocities of sediments). Mitteilungen Franzius Institute Univ. Hannover, Vol. 46, ISSN 0340 0077, full text for example at Researchgate).

However, this citing error is not at the expense of the authors. Leo v. Rijn, quoted this formula accidentally incorrectly. In others of his publications he has quoted this formula correctly.

Best wishes

Ulrich Zanke

Please also note the supplement to this comment:
https://www.hydrol-earth-syst-sci-discuss.net/hess-2018-511/hess-2018-511-SC1-supplement.pdf

---

## Referee Comment (RC1) · Anonymous Referee #1 · 4 Dec 2018

The paper focuses on the prognosis of suspended matter concentrations in rivers. For this purpose the authors use the program combination of TELEMAC and SISYPHE. Testing area is the Orne river in north-eastern France.

The authors state that TELEMAC/SISYPHE in its original version can only handle two sediment classes which is not sufficient in case of the natural grain size distributions. Therefore, the authors enhanced SISYPHE for handling 10 fractions. As a result, the simulated suspended sediment concentration (SSC) is markedly improved when the natural sediment mixture is represented by not only two but 10 fractions.

A special problem arises in the Orne River (and so in different other rivers) due to cohesive parts of bed sediments. The authors, on the one hand considered an erosion and transport behavior of the non cohesive sediments parts, not influenced by the cohesives, as long as the latter are less than 30%. On the other hand, beyond 50% cohesives the cohesive regime is assumed. It is recommended to explain this assignment in more detail.

However, a main message of this paper is that the TELEMAC / SISYPHE software is now being enhanced for use with 10 sediment fractions instead of just two. This is of interest especially to TELEMAC / SISYPHE users, but not to the wider community. The part describing TELEMAC / SISYPHE covers 1/3 of the paper. The second part leads to the conclusion that in the Orne River a simulation with 10 grain fractions at SSC gives a better result than just two fractions. That was expectable. It must also be noted that there are already other hydromorphodynamic software systems, eg, DELFT and MIKE (and others too) which have been multi-fractional for many years and work with a large number of soil layers. Also hydro-morphodynamic programs for especially fluid mud have been developed.

Before this background, it is recommended to shift the main content of the paper from an enhancement of the software to a better understandig of physical effects/processes. An example could be the interaction and erosion behavior of cohesive and cohesionless sediment within mixtures and their modeling. Also, a more deeply explanation why it is possible to linearly interpolate between the behavior of cohesionless sediment with that of cohesive parts (page 5, line 18).

———————————

---

## Referee Comment (RC2) · Anonymous Referee #2 · 19 Dec 2018

Review of "Sediment transport modelling in riverine environments: on the importance of grain-size distribution, sediment density and boundary conditions" by Lepesqueur et al.

This paper deals with improvements brought to the sediment transport module (SISYPHE) of the TELEMAC model by introducing multiple sediment sizes with specific densities, accounting for river bottom and banks contributions. The paper is well structured and easy to read. The introduction provides a relevant state of the art and clearly positions the study with respect to parent studies in the field of fine, short-term

hydromorphodynamic modelling. The modelling framework section mainly consists in describing the rationale of the coupling between TELEMAC and SISYPHE and ad hoc adaptations, without noticeable changes in the underlying physics of the models. The tested adaptations are put into practice on a single study site in moderate flow conditions, which although relevant seems a bit too limitative to explore all possibilities offered by the new formulations. The Results and discussion section is fair, most of the Figures and interpretations are convincing. I also rather agree with the conclusions and future scopes.

In my opinion this paper is worth publication for the improvements it describes to an already-validated and well-known modelling approach, even if this kind of contribution would have deserved a wider database of river contexts and stage conditions. In its present form it is more of a convincing feasibility study than a definitive proof. My second objection is that the 3D features of the model are rather "silenced" throughout the paper, at least not taken advantage of - maybe due to experimental difficulties (it is not easy to "measure" as many flow features as are predicted on these scales)

A few minor issues still need to be handled, listed in the following.

Title

"Boundary conditions" is not explicit enough for me - do the authors mean "upstream boundary conditions" or sediment availability on the bottom and river banks?

Abstract

P1L11 - I would rather say "the spatial pattern of particle distribution and density" instead of "particle site distribution and density" P1L13 - "rising and flood events" is a repetition P1L18 - It may seem somewhat tedious to only mention the upstream condition while a downstream control also exists, as Fr«1 most likely almost everywhere. However, this point is mentioned here and there in the paper and I don't know if it should be recalled/announced here.

Introduction

P2L19 - "erosion, transport and deposition" by chronological and phenomenological order (and also P3L13) P2L20 - I think the reader deserves a bit more tips on the reason why "those two parameters control the area where..." P2L25 - Could the authors provide additional indications regarding the conditions of the Durafour et al. study? P2L34 - "first" instead of "First"

Modelling framework

P3L14 - "which" instead of "and" before "allows" P3L26 - I thought $z_1$ was a "fictitious" horizontal level - its definition here is pretty unusual and the sense of "deeper vertical plane" is not straightforward at least to me. §2.4 - The (high) probability of flocculation for cohesive sediments is disregarded and only mentioned in the future scopes. Could the authors provide insights on the flow regimes in which flocculation can be ignored or will certainly occur, thus outlining the conditions of validity of the present approach? In complement, I think the $63\mu$m-limit is that between silt and the finest sand particles - this should also be mentioned. P4L25 - Value of M? Is ïĄťc0 ïĆź ïĄťcd and what are their typical values? P4L27 - Ws is not mentioned. P5L5 - "kinematic" instead of "cinematic" P5L17-19 - I think this section should be moved after the cases of cohesive and non-cohesive sediment have been described. P7L15-21 - Have you tried different upstream initial profiles or do you consider the one you chose is typical of pre-existing equilibrium conditions? If so, are the results only valid for such conditions?

Study Area...

P10L8-13 - These elements of discussion are fair and welcome but do you think the hypotheses assumed are strong hypotheses. Do you have any "independent" indications that your starting hypotheses are correct or are they just default hypotheses (or else, do the results drastically change if these assumptions prove wrong?)

Results and discussion

[Figure]

P12L26 - "The need for long simulations", does this hold for non-equilibrium initial conditions and if so, do you think it means the coupling is not strong or dynamic enough - as in quasi-static approximations for sediment movement, for example? P13L19 - Delete "slightly" P15L10 - From the point of view of physical processes at play, one may think that increased water stages and stream power would both dislodge and move heavier bottom particles and allow access to different sediment "sources" on the banks. Is it compatible with your approach or could it be described within it (for a stronger reconnection with experimental observations)?

Conclusion

P19L4-5 - It is unclear what boundary condition representation means

Future scope

P20L10-11 - It is not good practice to quote more than 5 references at once - please split the list and comment on the differences between studies and contexts.

---

## Author Comment (AC1) · 21 Jan 2019

*We would like first to thank Ulrich Zanke for the relevant remarks and comments:*

The paper was elaborated based by the use of SISYPHE. Within this code, the determination of settling velocities obviously is cited after v. Rijn (Hand book of Sediment Transport by currents and waves. Delft Hydraulics, Rep. H461, 1989).

Best wishes Ulrich Zanke

This formula, Eq. 6 in the paper under discussion has a spelling error. The number '18' in the denominator of the middle equation must be deleted. The area-wise subdivision is unnecessary (saves at most minimal computation time).

*We would like to thank Mr. Zanke for pointing out the error in equation 6 of our manuscript The necessary changes will be made in the final version of the manuscript.*

Apart from that, Eqn. 6 with v. Rijn cited as its author, is not correctly cited. Correct citation is Zanke, U. (1977): Berechnung der Sinkgeschwindigkeiten von Sedimenten (Determination of settling velocities of sediments). Mitteilungen Franzius Institute Univ. Hannover, Vol. 46, ISSN 0340 0077, full text for example at Researchgate). However, this citing error is not at the expense of the authors. Leo v. Rijn, quoted this formula accidentally incorrectly. In others of his publications he has quoted this formula correctly.

*We thank him for pointing out that equation 6 was not correctly quoted and we will correct this in the revised version of our manuscript.*

The middle equation also includes the solution "otherwise" and the solution for "d ≤10- 4". It should also be noted that the grain size data such as "d ≤10-4" only concern sands. However, the equation applies to any sediment densities in which the grain size assignments of Eqn. 6 then would not apply.

*In the SISYPHE module, the subroutine that computes fall rates is called only once per simulation (not at each time step). Consequently, this sub-division does not increase the computation time substantially.*

---

## Author Comment (AC3) · 21 Jan 2019

Review of "Sediment transport modelling in riverine environments: on the importance of grain-size distribution, sediment density and boundary conditions" by Lepesqueur et al. This paper deals with improvements brought to the sediment transport module (SISYPHE) of the TELEMAC model by introducing multiple sediment sizes with specific densities, accounting for river bottom and banks contributions. The paper is well structured and easy to read. The introduction provides a relevant state of the art and clearly positions the study with respect to parent studies in the field of fine, short-term

*We would like first to thanks the second reviewer for his careful reading of the paper and his pertinent remarks and comments.*

The modelling framework section mainly consists in describing the rationale of the coupling between TELEMAC and SISYPHE and ad hoc adaptations, without noticeable changes in the underlying physics of the models. The tested adaptations are put into practice on a single study site in moderate flow conditions, which although relevant seems a bit too limitative to explore all possibilities offered by the new formulations. The Results and discussion section is fair, most of the Figures and interpretations are convincing. I also rather agree with the conclusions and future scopes. In my opinion this paper is worth publication for the improvements it describes to an already-validated and well-known modelling approach, even if this kind of contribution would have deserved a wider database of river contexts and stage conditions. In its present form it is more of a convincing feasibility study than a definitive proof. My second objection is that the 3D features of the model are rather "silenced" throughout the paper, at least not taken advantage of - maybe due to experimental difficulties (it is not easy to "measure" as many flow features as are predicted on these scales) A few minor issues still need to be handled, listed in the following. Title "Boundary conditions" is not explicit enough for me - do the authors mean "upstream boundary conditions" or sediment availability on the bottom and river banks?

*We thank reviewer2 for this comment. We will change "boundary condition" by "upstream suspended sediment concentrations".*

Abstract P1L11 - I would rather say "the spatial pattern of particle distribution and density" instead of "particle site distribution and density"

*We thank reviewer2 for this comment. We will follow his recommendation.*

P1L13 - "rising and flood events" is a repetition

*We thank reviewer2 for this comment. We will modify the manuscript accordingly*

P1L18 - It may seem somewhat tedious to only mention the upstream condition while a downstream control also exists, as Fr«1 most likely almost everywhere. However, this point is mentioned here and there in the paper and I don't know if it should be recalled/announced here.

*Here, we tested the sensitivity to the way the upstream suspended sediment concentrations are defined (distributed on up to 10 size classes). Downstream, we compared the results of the different numerical simulations and the SSC measurements. We did not impose the downstream sediment concentrations in the model. However, due to the presence of the dam, the measure water depth is used as a downstream (flow) boundary condition. We will clarify these points in the revised version of the manuscript.*

Introduction

P2L19 - "erosion, transport and deposition" by chronological and phenomenological order (and also P3L13)

*We thank reviewer2 for this comment. We will follow this recommendation.*

P2L20 - I think the reader deserves a bit more tips on the reason why "those two parameters control the area where..."

*We thank reviewer2 for this comment. We changed the sentence as follow:*

*"The fall velocity controlled by sediment density and size determines the preferential deposit zones"*

P2L25 - Could the authors provide additional indications regarding the conditions of the Durafour et al. study?

*Durafour et al. (2014) compared bed load during tidal cycles and empirical formulations: they found that distributing bed load fluxes over a larger number of grain size classes significantly reduced differences between measurement and in situ observations. We will mention this in the revised version of our manuscript.*

P2L34 - "first" instead of "First" Modelling framework

*We thank reviewer2 for this comment. We will follow this recommendation.*

P3L14 - "which" instead of "and" before "allows"

*We thank reviewer2 for this comment. We will follow this recommendation.*

P3L26 - I thought z1 was a "fictitious" horizontal level - its definition here is pretty unusual and the sense of "deeper vertical plane" is not straightforward at least to me.

*We thank reviewer2 for this comment. We will change by "altitude of the first horizontal plane above the bottom".*

§2.4 - The (high) probability of flocculation for cohesive sediments is disregarded and only mentioned in the future scopes. Could the authors provide insights on the flow regimes in which flocculation can be ignored or will certainly occur, thus outlining the conditions of validity of the present approach?

*The flocculation process is well documented in estuary system, lake or deep ocean. In riverine environment (especially in small rivers like the Orne river) only a few authors discussed it and there is consequently a lack of literature with respect to flocculation modelling in this kind of environment. The shear rate which is one of the main physical parameter responsible for collision-aggregation-disaggregation is generally high in this kind of environment which do not allow the formation of macroflocs: the variation of fall velocity of cohesive sediment is hence limited compared to an estuary. The flocculation is not the only process that should be taken into account, indeed in riverine environment terrestrial aggregates, aquatic aggregates (eroded substratum) and flocs (aggregates formed in the water column) are also present and as a consequence the different aggregates do not have the same composition and properties (shear strength, density, fractal dimension).*

In complement, I think the 63μm-limit is that between silt and the finest sand particles - this should also be mentioned.

*We thank reviewer2 for this comment. we only mentioned in §2.4 the distinction of the properties (cohesion or not) and not the names of the grain size class. We changed our sentence as follow: "In SISYPHE, the distinction between cohesive (i.e., mud) and non-cohesive sediment is based on the sediment diameter: the sediment is considered cohesive below 63 μm (silts and clays) and non-cohesive beyond 63 μm."*

P4L25 - Value of M? Is ï ̧At'c0 ïC ´z ï ̧At'cd and what ´ are their typical values?

*The erosion constant M has been set to $2.4.10^{-5}$ kg/sm$^2$. Typical values are ranging from $10^{-5}$ to $5.10^{-3}$ kg/sm$^2$.*

*(We can not read the second question, we supposed it is referring to the shear stress and the critical shear stress)*

*The critical shear stress of the mud has been measured with the help of a scissometer: the critical shear strength of mud erosion was estimated at 0.48 Pa for the top layer and 0.84 Pa at 15 cm depth (linear interpolation is performed to attribute to each bottom's layer the appropriate critical shear stress). For consolidated sludge, the typical value of the critical shear stress would be between 0.3 and 6 Pa.*

P4L27 - Ws is not mentioned.

*We thank reviewer2 for this comment. We will mentioned this corresponds to the fall velocity.*

P5L5 - "kinematic" instead of "cinematic"

*We thank reviewer2 for this comment. We will follow this recommendation.*

P5L17-19 - I think this section should be moved after the cases of cohesive and non-cohesive sediment have been described.

*We thank reviewer2 for this comment. We will follow this recommendation*

P7L15-21 - Have you tried different upstream initial profiles or do you consider the one you chose is typical of pre-existing equilibrium conditions? If so, are the results only valid for such conditions? Study Area...

*The paragraph p7L15-21 concerns the strata of the numerical model: we have decomposed the layers of the bottom in the initial state according to the median diameter and we only tried this one.*

P10L8-13 - These elements of discussion are fair and welcome but do you think the hypotheses assumed are strong hypotheses. Do you have any "independent" indications that your starting hypotheses are correct or are they just default hypotheses (or else, do the results drastically change if these assumptions prove wrong?)

*The spatial distribution of sediment grain size in the river bed potentially plays an important role in the erosion and deposition fluxes (depending on the flow conditions). However, knowing the actual distribution of sediment grain size is not possible and this leads to local overestimation or underestimation of sediment fluxes. However, only limited variations were observed on the collected bottom sediment samples (four transects). We will add further explanations in the manuscript.*

Results and discussion

P12L26 - "The need for long simulations", does this hold for non-equilibrium initial conditions and if so, do you think it means the coupling is not strong or dynamic enough - as in quasi-static approximations for sediment movement, for example?

*In this context, the phrase "need for a long-term simulation" refers to the need for a numerical adjustment of the sediment distribution and the bathymetry at the beginning of the simulations in order to avoid unrealistic sediment fluxes..*

*The quasi-static approximation of the movement of sediments is indeed questionable, particularly in the case of sheet flow, fluid slime formation or small bedforms (eg ripples formation): another approach, such as modelling in two-phase flow, would be of course more appropriate, but also more time consuming. We will add a remark on this in the manuscript*

P13L19 - Delete "slightly"

*We thank reviewer2 for this comment. We will follow this recommendation.*

P15L10 - From the point of view of physical processes at play, one may think that increased water stages and stream power would both dislodge and move heavier bottom particles and allow access to different sediment "sources" on the banks. Is it compatible with your approach or could it be described within it (for a stronger reconnection with experimental observations)?

*The dynamic coupling of Telemac and Sysiphe is actually meant to allow simulations for higher/lower flow rates. We believe that the model would not need further development to simulate sediment transport during higher magnitude flood events. Of course, this is an interesting perspective to test the same model in such conditions, but we do not have the necessary validated dataset at that time.*

Conclusion

P19L4-5 - It is unclear what boundary condition representation means Future scope

*We thank reviewer2 for this comment. This sentence resume what we did in this study, hence the boundary condition there mean the SSC imposed at the upstream boundary condition: we will add the word "upstream SSC" in the sentence to make it clearer.*

P20L10-11 - It is not good practice to quote more than 5 references at once - please split the list and comment on the differences between studies and contexts.

*We thank reviewer2 for this comment. We will split the list of the quotations.*

---

## Author Response (AR1)

This document reports the answers to Editor's and Reviewers' comments. Editor's and Reviewers' comments are indicated in black front, corresponding answers in blue font and copy/paste from the revised manuscript as a support in green italic font.

We would like first to thank the two reviewers for their careful reading of the paper. We also would like to thank the editor and the two reviewers for their pertinent remarks and comments. In addition to the current answer to reviewers and editor we enclosed the revised version of our manuscript according to reviewers' recommendations. In the revised version of the manuscript, red characters indicate the changes that have been made.

Editor Decision: Reconsider after major revisions

After thoroughly reading your manuscript and I concur with the generally positive attitude of both reviewers. In line with their assessment I think that the manuscript needs to be revised and that the study should be based on more solid grounds. I noted that the recommendation of reviewer II (accept after technical corrections) is inconsistent with the following central statement of her/his assessment "In my opinion this paper is worth publication for the improvements it describes to an already-validated and well-known modelling approach, even if this kind of contribution would have deserved a wider database of river contexts and stage conditions". As this point should essentially be addressed in the new manuscript and I think this requires major revisions. Within those you should also shift the focus of the presentation to a stronger degree on the underlying physics, as recommended by reviewer I.

We thank Editor for these comments. It was also originally our intention to test the proposed approach on various flood event. However, due to technical constrain it was impossible to collect the necessary dataset. In the new version of the manuscript, according to Editor's and reviewers 'comments, we refocused the objective of the study.

Anonymous Referee #1

The paper focuses on the prognosis of suspended matter concentrations in rivers. For this purpose, the authors use the program combination of TELEMAC and SISYPHE. Testing area is the Orne river in north-eastern France. The authors state that TELEMAC/SISYPHE in its original version can only handle two sediment classes which is not sufficient in case of the natural grain size distributions. Therefore, the authors enhanced SISYPHE for handling 10 fractions. As a result, the simulated suspended sediment concentration (SSC) is markedly improved when the natural sediment mixture is represented by not only two but 10 fractions.

We thank reviewer 1 for these comments.

A special problem arises in the Orne River (and so in different other rivers) due to cohesive parts of bed sediments. The authors, on the one hand considered an erosion and transport behavior of the non cohesive sediments parts, not influenced by the cohesives, as long as the latter are less than 30%. On the other hand, beyond 50% cohesives the cohesive regime is assumed. It is recommended to explain this assignment in more detail. Also, a more deeply explanation why it is possible to linearly

interpolate between the behavior of cohesionless sediment with that of cohesive parts (page 5, line 18).

We thank reviewer 1 for this remark. We further explained this assignment in the revised version of our manuscript. The evaluation of erosion and deposition of the sediment mixture "cohesive and non-cohesive" we used is exactly the same as the official version of SISYPHE: it is based on the modelling framework previously proposed by Waeles (2005) inspired by Van Ledden (2002). This empirical approach to evaluate erosion fluxes of the sediment mixture was developed from experimental results (Van Ledden 2003, Mitchener and Torfs (1996), Panagiotopoulos 1997): these experiments clearly show the influence of the mud content on the threshold of movement of sandy bottom. In Panagiotopoulos (1997), for example, it was shown that the threshold for initiating sand motion is close to that of sand (between 0 to 30% of mud), increases substantially between 30 % to 50 % mud contents and finally reaches the mud's critical shear stress for mud content higher than 50%.

As highlighted by Reviewer 1, the main arbitrary choice made in SISYPHE (according to Waeles (2005) and Villaret (2010)) is to linearly interpolate between the erosion behavior of cohesionless sediment and that of cohesive one between 30% and 50% mud contents. Other authors ( Mitchener and Torfs (1996) and Jacobs et al. (2011)) suggested to apply cohesive sediment erosion behavior from 30% of mud on. However, one can argue that the linear interpolation may induce a smoother transition between cohesive and non-cohesive regimes. We agree that testing other approaches that the one implemented in SISYPHE would be of interest in general, but one can argue that other approaches implemented in hydromorphodynamic models are rather similar and we believe that it would go beyond the scope of our paper. Consequently, we decided to use the original equations implemented in SISYPHE to simulate sediment mixture erosion.

We added the following paragraph in the revised version of the manuscript:

*"According to the observations made by Panagiotopoulus (1997), the critical shear stress of sand depends on the mud fraction : with a mud fraction lower than 30%, the critical shear stress of sand is a little influenced by the mud content; whereas it reaches that of pure mud for mud fractions higher than 50%.*
*According to this, in SYSIPHE, the non-cohesive sediment is eroded as pure sand (non-cohesive regime) if the mass fraction of mud is below 30% and as mud (cohesive regime) if the mass fraction of mud is beyond 50% in the top layer of the river bottom sediment. Moreover, following Waeles (2005) and Villaret (2010), a linear interpolation between the two aforementioned formulations is used when the mud fraction is between 30% and 50%. One could argue that such linear interpolation is rather simplistic. For example, other authors (e.g., Mitchener and Torfs (1996) and Jacobs et al. (2011)) suggested applying cohesive erosion regime from 30% of mud on. However, a linear interpolation may induce a smoother transition between cohesive and non-cohesive regimes. Consequently, we decided to keep the original formulation implemented in SYSIPHE.*
*."*

However, a main message of this paper is that the TELEMAC / SISYPHE software is now being enhanced for use with 10 sediment fractions instead of just two. This is of interest especially to TELEMAC / SISYPHE users, but not to the wider community. The part describing TELEMAC / SISYPHE covers 1/3 of the paper. The second part leads to the conclusion that in the Orne River a simulation with 10 grain fractions at SSC gives a better result than just two fractions. That was expectable. It must also be noted that there are already other hydromorphodynamic software systems, eg, DELFT and MIKE (and others

too) which have been multi-fractional for many years and work with a large number of soil layers. Also hydro-morphodynamic programs for especially fluid mud have been developed.

Before this background, it is recommended to shift the main content of the paper from an enhancement of the software to a better understandig of physical effects/processes. An example could be the interaction and erosion behavior of cohesive and cohesionless sediment within mixtures and their modeling.

We thank reviewer 1 for this remark and we agree that the main objective of the paper is not to show a further development of Sisyphe only. We exposed the developments of SISYPHE in order to enable the understanding of the developments that have been carried out. As suggested by reviewer 1, the main objective of our paper is to evaluate how crucial it is to set up a hydromorphodynamic model using a realistic grain size distribution (based on in situ data). We consequently believe that our study is useful for a wider community than that of TELEMAC users. We better formulated the objective and the discussion according to this remark within the manuscript.

Anonymous Referee #2

Review of "Sediment transport modelling in riverine environments: on the importance of grain-size distribution, sediment density and boundary conditions" by Lepesqueur et al. This paper deals with improvements brought to the sediment transport module (SISYPHE) of the TELEMAC model by introducing multiple sediment sizes with specific densities, accounting for river bottom and banks contributions. The paper is well structured and easy to read. The introduction provides a relevant state of the art and clearly positions the study with respect to parent studies in the field of fine, short-term hydromorphodynamic modelling.

We would like first to thanks the second reviewer for his careful reading of the paper and his pertinent remarks and comments.

The modelling framework section mainly consists in describing the rationale of the coupling between TELEMAC and SISYPHE and ad hoc adaptations, without noticeable changes in the underlying physics of the models.

We thank reviewer2 for this comment.

The physical parameterization has been slightly modified compared to the official version of SISYPHE:
- we added the Smith and McLean formulation for the erosion flux of non-cohesive sediments;
- we added the Soulsby's formulation for the critical shear stress of non-cohesive sediments;

We have developed the SISYPHE model to allow:
- a particle size distribution of up to 10 classes (sediment mixture: cohesive and non cohesive)
- a different density for each class of sediment
- a different suspended sediment concentration for each sediment class imposed at the boundary conditions.

The tested adaptations are put into practice on a single study site in moderate flow conditions, which although relevant seems a bit too limitative to explore all possibilities offered by the new formulations. The Results and discussion section is fair, most of the Figures and interpretations are convincing. I also

rather agree with the conclusions and future scopes. In my opinion this paper is worth publication for the improvements it describes to an already-validated and well-known modelling approach, even if this kind of contribution would have deserved a wider database of river contexts and stage conditions. In its present form it is more of a convincing feasibility study than a definitive proof.

*We thank reviewer2 for this comment. Testing a model over various hydrodynamic conditions and apply it for different domain is always indeed more relevant. In our case, from a monitoring during three years we only got one complete dataset for the specific flood described in this paper. The dataset includes: - the upstream and downstream suspended sediment concentration, measurement by two autosampler and one turbiditymeter; - the sediment grain size distribution of the riverbed and the river's banks; - the flow rate and the water level. Unfortunately, autosampler's malfunctioning did not permit us to collect a complete dataset for another flood event having a reliable time series of suspended sediment concentration.*

*We presented results based on a higher magnitude flood event during the Intercoh (2017) and River Flow (2018) conferences. During this flood event suspended sediment concentration was only measured by the downstream turbiditymeter. As a consequence, we were forced (due to lack of data) to impose the model upstream boundary condition based on the downstream boundary measurement. In this context, it was not really possible to evaluate model results in detail.*

My second objection is that the 3D features of the model are rather "silenced" throughout the paper, at least not taken advantage of - maybe due to experimental difficulties (it is not easy to "measure" as many flow features as are predicted on these scales) A few minor issues still need to be handled, listed in the following.

*We thank reviewer2 for this comment. The maximum water level that we have in this domain is 4m at the downstream boundary due to the dam. For technical reasons, it was not possible to collect 3D measurements of suspended sediment concentration and we assumed a well-mixed water column.*

Title "Boundary conditions" is not explicit enough for me - do the authors mean "upstream boundary conditions" or sediment availability on the bottom and river banks?

*We thank reviewer2 for this comment. We changed the title as follows:*

*"Sediment transport modelling in riverine environments: on the importance of grain-size distribution, sediment density and suspended sediment concentrations at upstream boundary"*

Abstract P1L11 - I would rather say "the spatial pattern of particle distribution and density" instead of "particle site distribution and density" & P1L13 - "rising and flood events" is a repetition

*We thank reviewer2 for this comment. We have modified the sentence as follows:*

*"However, modelling exercises often neglect suspended sediment properties (e.g. sediment densities and grain-size distribution), even though such properties are known to directly control the sediment particle dynamics in the water column during flood events."*

P1L18 - It may seem somewhat tedious to only mention the upstream condition while a downstream control also exists, as Fr≪1 most likely almost everywhere. However, this point is mentioned here and there in the paper and I don't know if it should be recalled/announced here.

*Here, we tested the sensitivity to the way the upstream suspended sediment concentrations are defined (distributed on up to 10 size classes). Downstream, we compared the results of the different numerical simulations and the SSC measurements. We did not impose the downstream sediment concentrations in the model. However, due to the presence of the dam, the measure water depth is used as a downstream hydraulic boundary condition. We clarified these points in the revised version of the manuscript.*

C2 HESSD Interactive comment Printer-friendly version Discussion paper

Introduction

P2L19 - "erosion, transport and deposition" by chronological and phenomenological order (and also P3L13)

*We thank reviewer2 for this comment. We followed this recommendation. We have modified the sentences as follows:*

"This vertical differentiation of sediments complicates the modelling of sediment erosion, transport and deposition."

"Hydromorphodynamic models often simulate sediment dynamics according to three main processes, namely erosion, transport (via suspended load and bed load) and deposition."

P2L20 - I think the reader deserves a bit more tips on the reason why "those two parameters control the area where…"

*We thank reviewer2 for this comment. We changed the sentence as follows:*

"These two parameters therefore control the preferential deposit zones as particles with lower/higher fall velocity will be deposited in different areas."

P2L25 - Could the authors provide additional indications regarding the conditions of the Durafour et al. study?

*We thank reviewer2 for this comment. We added this sentence in the manuscript:*

"Durafour et al. (2014) compared various empirical formulations of bed load during tidal cycles and found that distributing bed load fluxes over a larger number of grain size classes significantly reduced differences between predictions and in situ observations."

P2L34 - "first" instead of "First" Modelling framework

*We thank reviewer2 for this comment. We followed this recommendation.*

P3L14 - "which" instead of "and" before "allows"

*We thank reviewer2 for this comment. We edited the text as follows:*

"Sediment transport is decoupled into the bed load and suspended load which allows sediment concentrations in the water column to be computed."

P3L26 - I thought $z_1$ was a "fictitious" horizontal level - its definition here is pretty unusual and the sense of "deeper vertical plane" is not straightforward at least to me.

*We thank reviewer2 for this comment. We changed for "altitude of the first horizontal plane above the bottom". We edited the text as follows:*

"In Eq. 1, $\rho$ is the water density, $u_*$ the friction velocity, $z_1$ the "altitude of the first horizontal plane above the bottom", $u_{z_1}$ the near bed flow velocity $\kappa = 0.4$ the von Kármán constant, $k_s \approx 2.5 d_{50}$ the Nikuradse bed roughness, and $d_{50}$ the median bottom sediment grain size."

§2.4 - The (high) probability of flocculation for cohesive sediments is disregarded and only mentioned in the future scopes. Could the authors provide insights on the flow regimes in which flocculation can be ignored or will certainly occur, thus outlining the conditions of validity of the present approach?

*The flocculation process is well documented in estuary system, lake or deep ocean. In riverine environment (especially in small rivers like the Orne river) only a few authors discussed it and there is consequently a lack of literature with respect to flocculation modelling in this kind of environment. The shear rate which is one of the main physical parameter responsible for collision-aggregation-disaggregation is generally high in this kind of environment which do not allow the formation of macroflocs: the variation of fall velocity of cohesive sediment is hence limited compared to an estuary.*

In complement, I think the 63µm-limit is that between silt and the finest sand particles - this should also be mentioned.

*We thank reviewer2 for this comment. we only mentioned in §2.4 the distinction of the properties (cohesive or non cohesive) and not the properties of the grain size class. We changed our sentence as follows:*

"In SISYPHE, the distinction between cohesive (i.e., mud) and non-cohesive sediment is based on the sediment diameter: the sediment is considered cohesive below 63 µm (silts and clays) and non-cohesive beyond 63 µm."

P4L25 - Value of M? Is ï ˛At'c0ïC ´z ï ˛At'cd and what´ are their typical values?

*The erosion constant M has been set to $2.4.10^{-5}$ kg/sm². Typical values are ranging from $10^{-5}$ to $5.10^{-3}$ kg/sm².*

*(We can not read the second question, we supposed it is referring to the shear stress and the critical shear stress)*

*The critical shear stress of the mud has been measured using a scissometer: the critical shear strength of mud erosion was estimated at 0.48 Pa for the top layer and 0.84 Pa at 15 cm depth (linear interpolation is performed to attribute to each bottom's layer the appropriate critical shear stress). For consolidated sludge, the typical value of the critical shear stress would be between 0.3 and 6 Pa.*

*We edited the text as follows:*

"In Eq. 4, M is the Partheniades constant set to $2.4.10^{-5}$ kg.s-1m-2, $\tau_0$ is the shear stress and $\tau_{ce}$ is the critical shear stress. The critical shear stress of the mud has been defined based on measurements using a scissometer: the critical shear strength of mud erosion was estimated at 0.48 Pa for the top layer and 0.84 Pa at 15 cm depth (a linear interpolation is used to attribute to each bottom layer an individual critical shear stress)."

P4L27 - Ws is not mentioned.

*We thank reviewer2 for this comment. We now mentioned this corresponds to the fall velocity.*

*We edited the text as follows:*

"In Eq. 5, $C$ is the suspended mud concentration in the water column, $\tau_{cd}$ the critical constraint of deposition (set at 0.001Pa) and $W_s$ is the fall velocity:"

P5L5 - "kinematic" instead of "cinematic"

*We thank reviewer2 for this comment. We followed this recommendation.*

P5L17-19 - I think this section should be moved after the cases of cohesive and non-cohesive sediment have been described.

*We thank reviewer2 for this comment. We followed this recommendation. We moved this paragraph just after the equation of the flux of deposition of non-cohesive sediment as this paragraph deals with the concept of reference concentration.*

P7L15-21 - Have you tried different upstream initial profiles or do you consider the one you chose is typical of pre-existing equilibrium conditions? If so, are the results only valid for such conditions? Study Area...

*The paragraph p7L15-21 concerns the strata of the numerical model: we have decomposed the layers of the bottom in the initial state according to the median diameter and we only tried this one.*

P10L8-13 - These elements of discussion are fair and welcome but do you think the hypotheses assumed are strong hypotheses. Do you have any "independent" indications that your starting hypotheses are correct or are they just default hypotheses (or else, do the results drastically change if these assumptions prove wrong?)

*The spatial distribution of sediment grain size in the river bed potentially plays an important role in the erosion and deposition fluxes (depending on the flow conditions). However, knowing the actual distribution of sediment grain size is not possible. Moreover, only limited variations were observed on the collected bottom sediment samples (four transects). The riverbed and the river banks both exhibited a rather homogeneous grain size distribution over the river section: a coarser one for the riverbed and a finer one for the banks.*

Results and discussion

P12L26 - "The need for long simulations", does this hold for non-equilibrium initial conditions and if so, do you think it means the coupling is not strong or dynamic enough - as in quasi-static approximations for sediment movement, for example?

*In this context, the expression "need for a long-term simulation" refers to the need for a numerical adjustment of the sediment distribution and the bathymetry at the beginning of the simulations in order to avoid unrealistic sediment fluxes. This is essential when the model configuration is established with an artificial sediment grain size distribution. For example, some authors use four classes of*

*sediment and set the fraction of each class to 25% at the beginning of the simulation: in this case, it is necessary to "warm up" the model for pre-initializing the sediment spatial distribution over the model domain and avoid unrealistic sediment fluxes at the beginning of the simulation.*

*The quasi-static approximation of the movement of sediments is indeed questionable, particularly in the case of sheet flow, fluid mud formation or small bedforms (e.g. ripples formation): another approach, such as modelling in two-phase flow, would be of course more appropriate, but also more time consuming.*

P13L19 - Delete "slightly"

*We thank reviewer2 for this comment. We followed this recommendation.*

P15L10 - From the point of view of physical processes at play, one may think that increased water stages and stream power would both dislodge and move heavier bottom particles and allow access to different sediment "sources" on the banks. Is it compatible with your approach or could it be described within it (for a stronger reconnection with experimental observations)?

*The dynamic coupling of Telemac and Sisyphe is actually meant to allow simulations for higher/lower flow rates. We believe that the model would not need further development to simulate sediment transport during higher magnitude flood events. Of course, this is an interesting perspective to test the same model in such conditions, but we were unfortunately unable to collected the necessary dataset.*

Conclusion

P19L4-5 - It is unclear what boundary condition representation means Future scope

*We thank reviewer2 for this comment. This sentence summarizes what we did in this study, hence the boundary condition there mean the SSC imposed at the upstream boundary: we will add the word "upstream SSC" in the sentence to make it clearer.*

*We edited the text as follows:*

"This study evaluates the influence of the sediment grain-size distribution, the sediment density and the upstream SSC representation on sediment transport/morphodynamic modelling."

P20L10-11 - It is not good practice to quote more than 5 references at once - please split the list and comment on the differences between studies and contexts.

*We thank reviewer2 for this comment. We modified the manuscript according to these recommendations.*

[revised manuscript text omitted]

---

## Referee Report (RR1)

**Referee comment to:**

Journal: HESS

Title: Sediment transport modelling in riverine environments: on the importance of grain-size distribution, sediment density and suspended sediment concentrations at upstream boundary

Author(s): J. Lepesqueur et al.

Hydrol. Earth Syst. Sci. Discuss., 2019, revised version

**General comments:**

This paper presents a hydromorphodynamic modelling study in a small river system, the Orne catchment in France. The objective of the study was to assess the importance of sediment characteristics (i.e. grain size distribution and distributed sediment densities) using the existing fully coupled hydromorphodynamic models TELEMAC 3D and SISYPHE. The latter allows for consideration of cohesive and non-cohesive sediment regimes and was further developed for the use of 10 grain size classes with varying densities for each class. In the framework of the modelling study, the sensitivity of SISYPHE to grain size distribution, sediment density and suspended sediment concentration at the upstream boundary was evaluated.

The modelling study in combination with the flood event data monitored at the Orne river and presented in this manuscript is well placed in HESS and worth publishing. In particular, with regard to the prediction of the resuspension and transport of particulate pollutants, it is necessary to consider several particle classes. However, I have some general comments regarding the configurations of the modelling study:

The standard configuration of the SISYPHE model allows for 2 grain size classes. This is clearly not enough for modelling suspended sediment concentrations. It is thus not surprising, that the model configuration with 10 size classes performs better than the configuration with only 2 classes. However, the 4 configurations (Page 12, Table 1) are a bit arbitrary chosen and too few tested possibilities for a sensitivity study.

In my opinion, it would be interesting, to find out, how many size classes are needed to receive a good prediction of the suspended sediment concentration and evolution of bathymetry. Are 10 classes necessary or can it be less? Therefore, the authors should consider testing also model configurations with other numbers of grain size classes.

Furthermore, the modelling study shows that the configuration with size specific sediment densities outperforms the configuration that uses the standard density of 2600 kg m$^{-3}$ for each class. This is also not surprising, since the average measured density for the sediments in the Orne river is 2300 kg m$^{-3}$ (Page 11, line 11). Therefore, a further model configuration with the measured average density of the sediments in the Orne should be modelled in order to see, if the distributed densities per size class are really needed or if an average measured value would also be adequate. The latter would also be less effort to measure than distributed densities for each class.

Some process representations of the SISYPHE model are simplified and pragmatic. This is adequate, since many sub-processes of erosion and deposition as well as the interaction of particles (in particular when cohesive sediments are involved) are too complex for a precise

physical description. Nevertheless, it is important to think about improvements of the model, as the authors have done in section 6, Future Scope. But, in my opinion, too many improvements are mentioned, which are unlikely to be achievable in the near future. I thus suggest, to mention only a few and feasible model adaptations.

In general, the Figures, in particular the cross comparisons in Figures 8 and 9, are informative and catchy. However, the overall presentation quality of the text could be improved. In the Results and Discussion section, many assumptions are made, which are not supported by observations or references in the literature. The discussion should be more precise. Furthermore, the present manuscript version contains many grammatical and typing errors. It should thus be thoroughly proofread.

**Specific comments and technical corrections:**

- Page 1, line 13: This study has a main objective to… → The main objective of this study is to…

- Page 1, line 16: allow → allows

- Page 1, line 21 and 24: insert 'configuration' behind 'model'

- Page 2, line 1: inputs → emissions

- Page 2, line 7: …of mineral particles of amorphous or poorly crystalline…. → a word is missing

- Page 2, line 16: 'transport formula' → better 'transport equation'

- Page 2, line 20 and 22 and also later in the manuscript: 'fall velocity' → better 'settling' or 'sink' velocity'

- Page 2, line 34: insert 'distributed' before 'sediment density'

- Page 3, line 2 and also very often later in the manuscript: SYSIPHE → SISYPHE

- Page 3, line 5: mad → made

- Page 3, line 12: 'This modelling framework has the following interests' → rephrase

- Page 4, line 12: deposit → deposition

- Page 6, line 10-21: I do not understand if the representation of deposition is the same for the cohesive and non-cohesive regime. Please clarify.

- Page 8, Figure 1: Pleas add a scale bar in the sub-figure on the right.

- Page 9, Figure 2: Please add the monitoring period and number of SSC measurements in the Figure caption.

- Page 10, Figure 4: Please add number of samples in the Figure caption.

- Page 11, line 8-11 and Figure 5: In Figure 5 the distributed densities for 10 grain size classes of the Orne river are displayed. How many samples were measured? Please consider to add error bars to show the variation of the sediment densities per grain size class. In addition, the high density of the 100 µm size class is interesting. Is there an explanation for that?

- Page 11, line 16-21: add dates of field campaigns.

- Page 12, Table 1: I suggest testing of additional model configurations (see general comments).

- Page 12, line 19: insert 'class' behind '100 μm'

- Page 12, line 23: delete 'obtained'

- Page 12, line 24: insert 'upstream' before 'boundary condition'

- Page 13, line 9: delete 'for the discussion'

- Page 13, line 16: underestimate → underestimates

- Page 13, line 12: increase → increased

- Page 13, line 21: move ''in the 10 CLD (2600 kg m$^{-3}$)' to line 20, between 'whereas' and 'we'

- Page 14, Figure 6: explain abbreviations in the Figure caption or refer to Table 1

- Page 15, line 10-11. This statement is not clear: What other kind of processes should influence the transport of suspended particles than advection and diffusion? Please clarify.

- Page 15, line 15: insert 'the' before '63 and …'

- Page 15, line 19-23: Please try to verify the assumptions in this paragraph.

- Page 16, Figure 7: it is very difficult to identify the grain size classes in the graph from the colors in the legend. Please use colors which are clearly differentiated

- Page 17, line 4: delete 'the' before 'erosion and…'

- Page 17, line 15: delete 'the' before 'deposition'

- Page 17, line 15-17: This statement is unclear. In addition, is there a reference in the literature?

- Page 19, line 18-19: This statement is imprecise. Please clarify.

- Page 20, line 11: ';;;;;' → missing reference?

- Page 20, line 5-21: In my opinion the list of improvements of the modelling framework is too comprehensive in the context of the manuscript. I thus recommend to focus on feasible improvements in the existing model framework.

---

## Author Response (AR2)

This document reports the answers to the Editor's and Reviewers' comments. The Editor's and Reviewers' comments are indicated in black front, corresponding answers in blue font and copy/paste from the revised manuscript (as a support) in red font.

We would like first to thank the Editor and two reviewers for their careful reading of the paper, their pertinent remarks and comments. In addition to the answer to the reviewers and the Editor, we enclosed the revised version of our manuscript according to reviewers' recommendations. In the revised version of the manuscript, red characters indicate the changes that have been made.

**General comments:**

This paper presents a hydromorphodynamic modeling study in a small river system, the Orne catchment in France. The objective of the study was to assess the importance of sediment characteristics (i.e. grain size distribution and distributed sediment densities) using the existing fully coupled hydromorphodynamic models TELEMAC 3D and SISYPHE. The latter allows for consideration of cohesive and non-cohesive sediment regimes and was further developed for the use of 10 grain size classes with varying densities for each class. In the framework of the modeling study, the sensitivity of SISYPHE to grain size distribution, sediment density and suspended sediment concentration at the upstream boundary was evaluated.

The modelling study in combination with the flood event data monitored at the Orne river and presented in this manuscript is well placed in HESS and worth publishing. In particular, with regard to the prediction of the resuspension and transport of particulate pollutants, it is necessary to consider several particle classes.

We thank reviewer 3 for these positive comments.

However, I have some general comments regarding the configurations of the modelling study:

The standard configuration of the SISYPHE model allows for 2 grain size classes. This is clearly not enough for modelling suspended sediment concentrations. It is thus not surprising, that the model configuration with 10 size classes performs better than the configuration with only 2 classes. However, the 4 configurations (Page 12, Table 1) are a bit arbitrary chosen and too few tested possibilities for a sensitivity study.

In my opinion, it would be interesting, to find out, how many size classes are needed to receive a good prediction of the suspended sediment concentration and evolution of bathymetry. Are 10 classes necessary or can it be less? Therefore, the authors should consider testing also model configurations with other numbers of grain size classes.

We thank Reviewer 3 for these comments and for raising these interesting points. We understand that choosing ten classes could be seen as slightly arbitrary. The number of classes was mainly chosen based on the vibratory sieve shaker at our disposal and on standard filters. As a matter of fact, for technical reasons, it is not straightforward to set up a model with a higher number of sediment classes. Moreover, as can be seen in the Figure 7 of the manuscript, the suspended load is limited to fine sediment (3 classes of diameter < 100µm). This figure shows that only the three finer classes are transported as suspended matter. Using fewer classes would then result in representing suspended load through two or one class only, which is closer to the scenario we set up with only two classes. Moreover, it is also important to highlighted that searching for an "optimal" number of class as suggested by Reviewer 3 would be really site and event specific. The "optimal" number of class therefore obtained would not be easily generalizable or scalable. It was also originally our intention

to test the proposed approach on various flood events. Unfortunately, during three years of monitoring, we were able to collect a complete validation dataset only for the proposed flood event. We argue that the objective of this paper is rather to provide some evidence of the benefit of increasing the number of sediment classes than identifying how many classes are best suited - as this number would be site and event specific.

Furthermore, the modelling study shows that the configuration with size specific sediment densities outperforms the configuration that uses the standard density of 2600 kg.m⁻³for each class. This is also not surprising, since the average measured density for the sediments in the Orne river is 2300 kg.m⁻³ (Page 11, line 11). Therefore, a further model configuration with the measured average density of the sediments in the Orne should be modelled in order to see, if the distributed densities per size class are really needed or if an average measured value would also be adequate. The latter would also be less effort to measure than distributed densities for each class.

We thank Reviewer 3for these comments.

We choose to consider an average value of 2600 kg.m⁻³ in the homogenous density scenario as this is the value commonly used in sediment transport modelling studies.
To follow reviewer 3 suggestion, we carried out a new simulation with a set up identical to the 10CLD configuration (referred below as 10CLD2600) using this time a sediment density of 2300 kg.m⁻³ (average value in the Orne sediment; 10CLD2300).
The scatter plots of the evolution and the D50 variation of this new simulation are shown below.

[Figure]

Cross-comparison of simulated bed elevation evolutions (elevation final-initial) for configurations 10CLD2300, 10CLD2600 and 10CL.

[Figure]

Cross-comparison of simulated bed D50 evolutions (final-initial) for configurations 10CLD2300, and 10CLD2600 and 10CL.

The left hand side panels show the scatter plots allowing the comparison of the 10CL and the 10CLD ($\rho$=2300 kg.m$^{-3}$) configurations.

The centre panels show the scatter plots allowing the comparison of the 10CL and the 10CLD ($\rho$=2600 kg.m$^{-3}$) configurations.

The right hand side panels show the scatter plots allowing the comparison of the 10CLD($\rho$=2600 kg.m$^{-3}$) and the 10CLD($\rho$=2300 kg.m$^{-3}$) configurations.

As can be seen in this figures the two simulations with a constant sediment density yield similar results. Moreover, the differences between the 10CLDs and the 10CL simulations are more significant that the difference between the two 10CLD simulations. This thus indicates that imposing a nominal density for each class has an influence on the model results and is therefore worth. Defining per-class sediment densities is especially relevant when simulating flood events with a wide range of magnitudes: as the flow intensity increases, larger sediment classes are transported. As a matter of fact, using class-based sediment densities helps in improving model results. In our experiment, the 100μm and the 65μm classes have respectively a density of 2850 and 1750 kg.m$^{-3}$. Using the mean density (2300 kg/m³) for both classes necessarily introduces a source of error in the model that can subsequently induce errors in the assessment of locations where fine sand and silt are deposited. We added the following sentences in the article to clarify this:

"In this sediment transport modelling study, we chose to consider an average density value of 2600 kg.m$^{-3}$ in the 10CLD scenario as this is the most commonly used value. One could argue that the average measured sediment density could also perform satisfactorily. To evaluate this option, we carried out an additional simulation identical to 10CLD but using a sediment density of 2300 kg.m$^{-3}$ (results not shown). The results obtained were similar to those obtained with the 10CLD configuration and different from the 10CL simulations, showing the added value of using nominal measured densities. This is arguably mainly due to the fact that only fine sediment classes are transported during this flood event and fine sediment classes have different nominal densities than the tested average values, namely 2300 and 2600 kg.m$^{-3}$ (see Fig. 5)."

Some process representations of the SISYPHE model are simplified and pragmatic. This is adequate, since many sub-processes of erosion and deposition as well as the interaction of particles (in particular when cohesive sediments are involved) are too complex for a precise physical description. Nevertheless, it is important to think about improvements of the model, as the authors have done in section 6, Future Scope. But, in my opinion, too many improvements are mentioned, which are unlikely to be achievable in the near future. I thus suggest, to mention only a few and feasible model adaptations.

We thank Reviewer 3 for these comments. We modified the future scope to take this into account. Especially, we clearly separate the improvements that can be envisaged in a near future from the others:

"The proposed sediment transport modelling framework is found to improve the accuracy of the results. However, additional developments could be considered in order to integrate bio-physico-chemical processes. Indeed, the temporal variability of the bio-physico-chemical conditions in rivers plays a key role in shaping the sediment dynamics during flood events. In this context, we envisage implementing two important developments:

1. A new generation of high-frequency measurement sensors could be used to record the model input data. A LISST sensor (Fugate, D. C. and Friedrichs, 2002), which measures the size and concentration of particles suspended in water, or a combination of two acoustic doppler current profilers (Jourdin et al., 2014) could for example be used to monitor the SSC for each individual sediment class. This would provide more realistic model inputs and more accurate validation data at the same time.

2. Flocculation processes could be integrated as they play a key role in sediment transport due to the fact that the density and the shape of flocs differ from those of individual sediment particles. As a result, their displacement in the water column is different from that of isolated sediment particles (Parker, 1972; Van der Lee, 2009). The integration of flocculation process could be implemented by coupling a morphodynamic model with a floc population model such as FLOCMOD (Verney et al., 2009, Lepesqueur et al., 2018).

In terms of longer-term developments, the erosion and deposition laws used in the morphodynamic model should also take into account interactions between sediment classes, as argued for example by Starck (2014). Indeed, many existing studies highlight the importance of the compaction of non-cohesive sediment (Swidersky, 1976), armouring (Egiazaroff, 1965), hiding/exposure (Ashida, 1973), filtration of fine particles by coarser sediment (Karim, 1982; Brunke, 1999; Herzig et al., 1970) and lubrication (Barry, 2006), together with biological processes (e.g. Arthur et al., 1980; Widdows et al., 2000; Le Hir et al., 2007). "

In general, the Figures, in particular the cross comparisons in Figures 8 and 9, are informative and catchy.
We thank Reviewer 3 for this positive comment.

However, the overall presentation quality of the text could be improved. In the Results and Discussion section, many assumptions are made, which are not supported by observations or references in the literature. The discussion should be more precise. Furthermore, the present manuscript version contains many grammatical and typing errors. It should thus be thoroughly proofread.

We thank Reviewer 3 for these remarks. We carefully proofread the manuscript with the help of a native English speaker. Moreover, we put some efforts on being more precise in the result and discussion part.

**Specific comments and technical corrections:**

Page 1, line 13: This study has a main objective to... The main objective of this study is to…
This has been corrected in the new version of the manuscript

Page 1, line 16: allow allows
We corrected the sentence.

Page 1, line 21 and 24: insert 'configuration' behind 'model'
We corrected the sentence.

Page 2, line 1: inputs emissions
We corrected the sentence

Page 2, line 7: ...of mineral particles of amorphous or poorly crystalline....  a word is missing
We corrected the sentence as follows:
"River sediments are heterogeneous aggregates, composite structures composed of amorphous or poorly crystalline mineral particles, organic matter, and biological matter (biofilms, bacteria, virus and bio-macromolecules)."

Page 2, line 16: 'transport formula' better 'transport equation'
This has been changed.

Page 2, line 20 and 22 and also later in the manuscript: 'fall velocity' better 'settling' or 'sink' velocity'
To our knowledge, both expressions are widely used in the literature. We did not really see the benefit of using systematically "settling" instead of "fall". As a consequence, we use both term in the manuscript.

Page 2, line 34: insert 'distributed' before 'sediment density'
We corrected this.

Page 3, line 2 and also very often later in the manuscript: SYSIPHE SISYPHE.
We corrected the typos.

Page 3, line 5: mad made
We corrected the typo.

Page 3, line 12: 'This modeling framework has the following interests' rephrase
We rephrased this sentence:

"We adopted this modelling framework for two main reasons:"

Page 4, line 12: deposit deposition
Corrected

Page 6, line 10-21: I do not understand if the representation of deposition is the same for the cohesive and non-cohesive regime. Please clarify.

For the non-cohesive sediment, deposition is always represented via eq 8:
"the deposition rate of the non-cohesive sediment is invariably computed using:
$$D = W_s * C_{ref} \tag{8}$$
In Eq. 8, D is the deposition rate and C_ref the reference sediment concentration at the bottom of the water column."

For the cohesive sediment, deposition is always represented via eq 5.
To clarify this point we removed the word deposition in the following sentence:

"Depending on the mud fraction (i.e., ratio between mud and total sediment mass) in the top layer of the river bed sediment, SISYPHE treats non-cohesive sediment erosion according to the so-called non-cohesive and cohesive regimes."

Page 8, Figure 1: Pleas add a scale bar in the sub-figure on the right.
We added a scalebar in Fig. 1

Page 9, Figure 2: Please add the monitoring period and number of SSC measurements in the Figure caption.
We added this information.

Page 10, Figure 4: Please add number of samples in the Figure caption.
We added this information

Page 11, line 8-11 and Figure 5: In Figure 5 the distributed densities for 10 grain size classes of the Orne river are displayed. How many samples were measured? Please consider to add error bars to show the variation of the sediment densities per grain size class.

Unfortunately, we did not repeat the density measurements many times and we are consequently not able to add error bars to show the variance of the density distributions between particle sizes. Moreover, we do not have the sample anymore. As a matter of fact, it is at that time impossible for us to answer the reviewer request. We apologize for this.

In addition, the high density of the 100 μm size class is interesting. Is there an explanation for that? We thank Reviewer 3 for these comments. Depending on the mineralogical aspect the density can be spread from 1400 kg.m$^{-3}$ (Morraine for example) up to 7600 kg.m$^{-3}$ (the exception of the Galena). The Schist or the Gneiss for example can have such density values (i.e. 2850 kg.m$^{-3}$). In the Orne River, sediment is not only composed of quartztite (2600 kg.m$^{-3}$) and can be attributed to soil microaggregates and residues of anthropogenic past and actual activities, which can explain the rather large density variability.

Page 11, line 16-21: add dates of field campaigns.
We add the dates of the field campaigns.

Page 12, Table 1: I suggest testing of additional model configurations (see general comments).
Please see the answers to the general reviewers comments

Page 12, line 19: insert 'class' behind '100 μm'
We did so in the new version of the manuscript.

Page 12, line 23: delete 'obtained'
We did so in the new version of the manuscript.

Page 12, line 24: insert 'upstream' before 'boundary condition'
We did so in the new version of the manuscript.

Page 13, line 9: delete 'for the discussion'
We did so in the new version of the manuscript.

Page 13, line 16: underestimate  underestimates
We did so in the new version of the manuscript.

Page 13, line 12: increase  increased
We did so in the new version of the manuscript.

Page 13, line 21: move ''in the 10 CLD (2600 kg m -3 )' to line 20, between 'whereas' and 'we'
We did so in the new version of the manuscript.

Page 14, Figure 6: explain abbreviations in the Figure caption or refer to Table 1
We will refer to Table 1 and explain abbreviations.

Page 15, line 10-11. This statement is not clear: What other kind of processes should influence the transport of suspended particles than advection and diffusion? Please clarify.
To clarify this statement, we edited the text as follow:
"Fig. 3 shows that the SSC time series have similar shapes and magnitudes at the upstream and downstream boundaries of the model. This indicates that erosion plays a limited role in the overall

sediment transport budget when compared to advection and dispersion, during this rather low magnitude flood event."

Page 15, line 15: insert 'the' before '63 and ...'
We did so in the new version of the manuscript.

Page 15, line 19-23: Please try to verify the assumptions in this paragraph.
We thank Reviewer 3 for this comments. This is actually more than an assumption.
When simulating high magnitude flood events, the flow velocities increase resulting in a larger number of sediment classes potentially eroded, transported and deposited. The studied flood event was of moderate magnitude and not all the 10 classes of sediment where transported in suspension. Only the five smallest classes were imposed as suspended sediments at the upstream boundary and only the four smallest classes were simulated in suspension at the dowstream boundrary. Indeed, the size of particles transported in suspension, especially through advection-dispersion processes, would be larger during events with higher flood magnitude. Moreover, it would increase the deposition and erosion of coarser particles along the river section.
The related sentences have been modified in the revised version of the manuscript:
"It is also worth mentioning that larger differences between the two configurations are expected in terms of simulated SSC for higher-magnitude flood events. Indeed, for such flood events, larger sediment particles are transported as a result of higher flow velocities. Distributing the upstream SSC over various sediment classes would then allow the transport of larger sediment particles via advection-dispersion in configuration 10CL."

Page 16, Figure 7: it is very difficult to identify the grain size classes in the graph from the colors in the legend. Please use colors which are clearly differentiated
We changed the color scale as suggested by the reviewer.

Page 17, line 4: delete 'the' before 'erosion and...'
We did so in the new version of the manuscript.

Page 17, line 15: delete 'the' before 'deposition'
We did so in the new version of the manuscript.

Page 17, line 15-17: This statement is unclear. In addition, is there a reference in the literature?
The related sentences have been modified in the revised version of the manuscript to clarify this statement:
"Moreover, we argue that the influence of sediment density on model simulations would be larger when simulating higher-magnitude flood events as the range of transported sediment sizes would be broader. Indeed, during larger flood events, we might expect that coarser sediments are transported, eroded and deposited. Moreover, a change in sediment density is associated with a change in fall velocity, which implies changes in the transport processes: a higher density reduces transport and, on the contrary, a lower density increases it. Changes in density would therefore also result in the displacement of erosion and deposition areas for coarser sediment, making bathymetry evolutions more markedly different in 10CL and 10CLD configurations during higher-magnitude flood events."

Page 19, line 18-19: This statement is imprecise. Please clarify.
The cross comparison of the final riverbed elevation evolution is a proxy of changes in location and amplitude of deposition and erosion.
The related sentence has been modified in the revised version of the manuscript to clarify this statement:
"Our analysis, based on a correlation of riverbed evolutions, also shows that using measured sediment densities instead of standard ones slightly changes the areas of erosion/deposition."

Page 20, line 11: ';;;;' missing reference?
Sorry for this error. This has been corrected in the new version of the manuscript

Page 20, line 5-21: In my opinion the list of improvements of the modeling framework is too comprehensive in the context of the manuscript. I thus recommend to focus on feasible improvements in the existing model framework.
We thank Reviewer 3 for this suggestion. We now separate improvements that can be done in a near future and longer-term perspectives in sediment transport modelling:
"The proposed sediment transport modelling framework is found to improve the accuracy of the results. However, additional developments could be considered in order to integrate bio-physico-chemical processes. Indeed, the temporal variability of the bio-physico-chemical conditions in rivers plays a key role in shaping the sediment dynamics during flood events. In this context, we envisage implementing two important developments:

1. A new generation of high-frequency measurement sensors could be used to record the model input data. A LISST sensor (Fugate, D. C. and Friedrichs, 2002), which measures the size and concentration of particles suspended in water, or a combination of two acoustic doppler current profilers (Jourdin et al., 2014) could for example be used to monitor the SSC for each individual sediment class. This would provide more realistic model inputs and more accurate validation data at the same time.

2. Flocculation processes could be integrated as they play a key role in sediment transport due to the fact that the density and the shape of flocs differ from those of individual sediment particles. As a result, their displacement in the water column is different from that of isolated sediment particles (Parker, 1972; Van der Lee, 2009). The integration of flocculation process could be implemented by coupling a morphodynamic model with a floc population model such as FLOCMOD (Verney et al., 2009, Lepesqueur et al., 2018).

In terms of longer-term developments, the erosion and deposition laws used in the morphodynamic model should also take into account interactions between sediment classes, as argued for example by Starck (2014). Indeed, many existing studies highlight the importance of the compaction of non-cohesive sediment (Swidersky, 1976), armouring (Egiazaroff, 1965), hiding/exposure (Ashida, 1973), filtration of fine particles by coarser sediment (Karim, 1982; Brunke, 1999; Herzig et al., 1970) and lubrication (Barry, 2006), together with biological processes (e.g. Arthur et al., 1980; Widdows et al., 2000; Le Hir et al., 2007)."